

**Enigmatic Fe-Mn-fueled Anaerobic Oxidation of Methane in sulfidic coastal**
**sediments of the Eastern Arabian Sea**
**Kalyani Sivan[1,2], Aditya Peketi[1,2], Aninda Mazumdar[1,2], Anjali Zatale[1,2], Sai Pavan**
**Kumar Pillutla[2,3], Ankita Ghosh[1,2], Mohd Sadique[2,3], Jittu Mathai[2]**
[1]*Academy of Scientific and Innovative Research (AcSIR), Ghaziabad-201002, India*
[2]*Gas hydrate Research Group, CSIR-National Institute of Oceanography, Dona Paula,*
*Goa-403004, India*
[3]*School of Earth, Ocean, and Atmospheric Sciences, Goa University, Taleigao Plateau,*
*Goa-403206, India*
*Correspondence to*:  Dr. Aninda Mazumdar, email: (maninda@nio.org)
**Abstract**
Anaerobic oxidation of methane (AOM) coupled with Fe-Mn reduction (Fe-Mn-AOM) is
considered a globally important biogeochemical process in marine sediments in addition to
sulfate-driven AOM ($SO_4^{2-}$-AOM) responsible for the consumption of methane, a strong
greenhouse gas. Most existing studies have emphasized the significance of Fe-Mn-AOM
activities in sediments below the depth of the sulfate methane transition zone (SMTZ) with
insignificant dissolved sulfide and sulfate concentrations in the porewaters. Here, we report for
the first time enigmatic geochemical evidence of focused Fe-Mn-AOM activity across the
SMTZ in the presence of high dissolved sulfide concentrations in a sediment core collected
within the seasonal coastal hypoxic zone of the Eastern Arabian Sea (West coast of India
(WCI)). The Fe-Mn-AOM activity is evident from the concurrent decrease in $CH_4$
concentrations, $\delta^{13}C_{CH4}$ and $\delta^{13}C_{DIC}$ values coupled with the enrichment of porewater $Fe^{2+}$ and





$Mn^{2+}$ concentrations at multiple depths below the seafloor. Since neither $CH_4$ nor reactive Fe
appears to be the limiting factor controlling the Fe-Mn-AOM activity, we hypothesize that the
focused Fe-Mn-AOM at multiple depths is likely fueled by the localization of metal-reducing
and methanotrophic microbial communities, leading to biogeochemical heterogeneity in a
dynamic seasonally hypoxic coastal environment sensitive to climate change. This study
highlights new insight into $CH_4$-S-Fe-Mn biogeochemical cycling with far-reaching
implications in climate studies linked to the estimation of sedimentary methane production and
consumption.

## 1 Introduction

The metal-driven anaerobic oxidation of methane (AOM) is a vital biogeochemical process
and may have a significant impact on global biogeochemical cycles, particularly in relation to
coupled $CH_4$-Fe-Mn-S cycling (Slomp et al., 2013; Riedinger et al., 2014; Leu et al., 2020).
Microbial Fe-Mn-driven AOM contributes to the global consumption of $CH_4$ in marine
sediments in addition to sulfate ($SO_4^{2-}$), nitrite ($NO_2^-$), and nitrate ($NO_3^-$)-driven AOM
(Raghoebarsing et al., 2006; Beal et al., 2009; Knittel et al., 2009; Ettwig et al., 2010; Riedinger
et al., 2014; Xiao et al., 2023). A potential syntrophic coupling of AOM and reduction of Fe-
Mn-(oxyhydr)oxides in marine sediments was first inferred from incubation experiments using
$^{13}$C-labeled $CH_4$ (Beal et al., 2009). The coupling of AOM and Fe-Mn-reduction is potentially
attributed to the activities of methanotrophs (ANME-1, ANME-3) and metal-reducing bacteria
(Beal et al., 2009; Oni et al., 2015), while Fe-Mn-(oxyhydr)oxide reduction may also be carried
out by ANME-2a, ANME-2c, and ANME-2d using multiheme cytochromes without syntrophic
metal-reducing bacterial partners (Ettwig et al., 2016; Scheller et al., 2016; Cai et al., 2018;
Leu et al., 2020). Plausible electron transport pathways coupling AOM and metal oxide
reduction include (a) direct electron transfer via microbial contact; (b) indirect electron transfer
by electron shuttling; (c) indirect electron transfer by a metal chelate; and (d) direct electron



transfer by nanowires (Folgosa et al., 2015; Wegener et al., 2015; He et al., 2018). The
biogeochemical reactions during Fe-Mn-AOM (Eq. 1 and 2) result in the mutual consumption
of $CH_4$ and Fe-Mn-(oxyhydr)oxides coupled with the enrichment of $Fe^{2+}$ and $Mn^{2+}$ in the
interstitial waters (Beal et al., 2009; Riedinger et al., 2014). Although $SO_4^{2-}$-AOM (Eq. 3) is
responsible for > 90% of global methane consumption, Fe-Mn-AOM is thermodynamically
more favorable than $SO_4^{2-}$-AOM (Beal et al., 2009) as evident from the $\triangle G$ values (Eq. 1, 2
and 3). The global predominance of $SO_4^{2-}$-AOM may be attributed to its kinetics (Lovley and
Phillips, 1987).
$CH_4 + 8\ Fe(OH)_3 + 15H^+ \rightarrow HCO_3^- + 8\ Fe^{2+} + 21H_2O$   $\triangle G$= -270.3 kJ mol$^{-1}$ $CH_4$      (1)
$CH_4 + 4\ MnO_2 + 7H^+ \rightarrow HCO_3^- + 4\ Mn^{2+} + 5H_2O$        $\triangle G$= -556 kJ mol$^{-1}$ $CH_4$        (2)
$CH_4 + SO_4^{2-} \rightarrow HS^- + HCO_3^- + H_2O$                                $\triangle G$= -10 to -40 kJ mol$^{-1}$ $CH_4$  (3)
The availability of reactive Fe-Mn-(oxyhydr)oxides and low to undetectable $SO_4^{2-}$
concentrations are considered essential geochemical conditions to drive Fe-Mn-AOM (Beal et
al., 2009; Oni et al., 2015; Aromokeye et al., 2019), however, a recent report (Li et al., 2019)
has shown the feasibility of Fe-Mn-AOM in $SO_4^{2-}$-rich porewaters. The kinetics of Fe-Mn-
AOM depend on the content and composition of particulate Fe-Mn-(oxyhydr)oxides and
soluble ferric-organo complexes (e.g., ferric citrate) (Ettwig et al., 2016). It is observed that the
soluble metal complexes and ferrihydrite have higher bioavailability and AOM activity than
other solid metal-(oxyhydr)oxides (Lovley and Phillips, 1987; Norði et al., 2013; Ettwig et al.,
2016; Scheller et al., 2016). However, several incubation experiments and cultures have shown
that crystalline goethite, hematite, and magnetite can also serve as significant electron
acceptors for Fe-AOM (Bar-Or et al., 2017; Aromokeye et al., 2019; Li et al., 2021).
The global distribution of Fe-Mn-AOM is plotted in Figure 1a. Here, we present the first tell-
tale evidence of Fe-Mn-AOM from a seasonally hypoxic shelf zone. The study was carried out



in a sediment core (SSD070/7/GC6) collected off the west coast of India (WCI) at a water
depth of 28.5 m (Figure 1b). The seasonal hypoxic zone off the WCI covers an area of 1,80,000
km$^2$ and is the largest of all coastal hypoxic systems (Naqvi et al., 2000). The hypoxia is
attributed to the upwelling of cold, nutrient-rich waters onto the shelf region during July to
September, associated with the southwest monsoon. During the southwest monsoon, the WCI
receives substantial terrestrial as well as marine organic matter due to enhanced land runoff
and upwelling-driven marine productivity respectively. In contrast, during northeast monsoon,
the WCI experiences downwelling from November to April resulting in the prevalence of oxic
and oligotrophic conditions in the water column (Naqvi et al., 2000; Schott et al., 2001).
Consequently, the region exhibits contrasting seasonal biogeochemical conditions including
hydrographic and depositional features (Naqvi et al., 2006; Mazumdar et al., 2012). In the
present study, we have investigated possible driving factors fueling Fe-Mn-AOM activity in a
seasonally hypoxic shelf zone characterized by contrasting redox and depositional conditions.

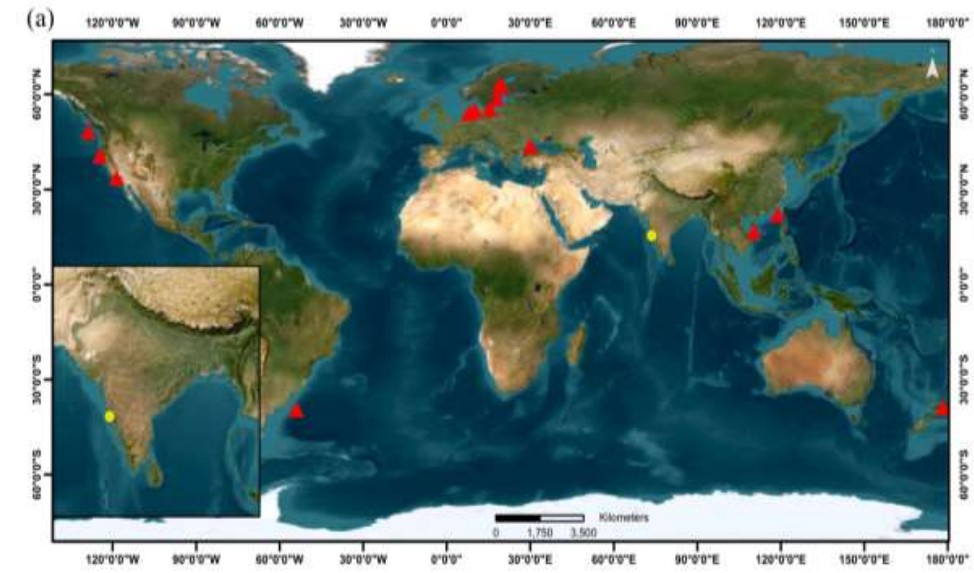




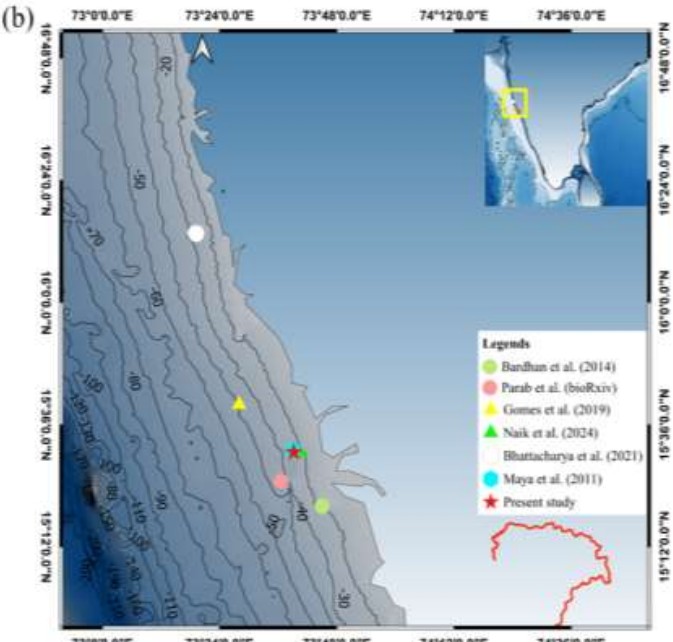


**Figure 1**. a) Global map showing various study locations (red-filled triangles) with reported Fe-Mn-

AOM activities. The yellow-filled circle represents the present study location. The references are listed

in the supplementary material. (b) Map showing locations of the present study (SSD070/7/GC6) along

with those collected in previous studies (Maya et al., 2011; Bardhan et al., 2014; Gomes et al., 2019;

Bhattacharya et al., 2021; Naik et al., 2023; Parab et al., *bioRxiv*) from the inner shelf off the West Coast

of India (WCI). The red star represents the present study location (SSD070/7/GC6), while other colored

symbols represent locations in the WCI where $\delta^{13}C$ and $\delta^{15}N$ of suspended particulate organic matter

and microbiological studies have been conducted.







## 2 Methodology

### 2.1 Onboard Sampling

The studied gravity core (SSD070/7/GC6, 5.25 m long) was collected onboard ORV *Sindhu Sadhana* in February 2020. The coring operation was carried out off Goa (latitude: 15.50985°N, longitude: 73.65313°E) from the seasonal hypoxic coastal zone of the Eastern Arabian Sea. Immediately after recovery, the core was subsampled onboard for hydrocarbon gas analysis and porewater extraction.

For headspace methane analysis, the sediment was extracted using 50 ml cut syringes at an interval of 10 cm and transferred into 20 ml headspace vials filled with 3 ml of KOH and 3 ml of $NaN_3$ to trap $CO_2$ and arrest microbial activities respectively. The vials were flushed with helium, homogenized, and stored at 4°C after sealing with butyl rubber septa. The core was subsampled onboard for porewater extraction at 10 cm intervals and transferred into 50 ml tarsons centrifuge vials using a 50 ml cut syringe under a stream of argon gas to avoid oxidation of dissolved sulfide. The samples were centrifuged at 7000 rpm for 20 minutes in a Remi C-30 centrifuge. The supernatant porewater was filtered through a 0.2 μm Whatman syringe filter. The filtered porewater was then stored in crimp vials under a helium head and subsequently preserved at 4°C for measurement of various constituents. For dissolved sulfide ($\Sigma HS^- = H_2S + HS^- + S^{2-}$) analysis, 1 ml of 1 M $CdNO_3$ was added to fix all sulfides present as CdS. Porewater samples for trace metal analysis were acidified with a known amount of concentrated supra-pure $HNO_3$. All porewater samples were immediately crimp-sealed under a helium head and stored at 4 °C until shore-based analysis. The core was subsampled for solid phase analysis at 1 cm resolution and stored at 4 °C for further analysis.



**2.2 Porewater and headspace gas analysis**

The headspace hydrocarbon gas concentrations were determined using a trace GC equipped with a flame ionization detector. Methane concentration measurements were carried out using an external calibration line prepared from six different gas mixtures ($CH_4$, $C_2H_6$, $C_3H_8$, $C_4H_{10}$, iso-$C_4H_{10}$, and $CO_2$) standards with concentrations varying from 10-1000 ppm. The carbon isotopic composition of methane was determined using an isotope ratio mass spectrometer (Thermo-Delta V plus) coupled with a gas chromatograph (Thermo-Trace GC Ultra). The carbon isotope ratios of methane ($\delta^{13}C_{CH4}$) are reported in delta notation as permil deviation from Vienna Pee Dee Belemnite (VPDB) standard, and the external precision calculated for the measurement is typically 0.07–0.09‰. The concentration of dissolved inorganic carbon (DIC) was measured using a carbon coulometer (UIC-CM5130) with ultrapure $CaCO_3$ as standard. The measurement yielded a sample reproducibility of $\pm$ 0.3 %. The carbon isotope ratios of DIC ($\delta^{13}C_{DIC}$) were analyzed using a Thermo Delta V continuous flow isotope ratio mass spectrometer coupled with a GASBENCH II equipped with a PAL auto-sampler, following Torres et al. (2005) and Peketi et al. (2020). Reference materials NBS-18, NBS-19, Carrara marble standards, and in-house laboratory standards (MERCK high purity $CaCO_3$ and Carbonate reference standard) were measured multiple times during the analysis with each batch. The carbon isotope composition of DIC is reported in delta notation as permil deviation from the VPDB standard. The measurement yielded a sample reproducibility of $\pm$ 0.02 ‰. Porewater samples were diluted 40-fold for the determination of trace metals using high-resolution inductively coupled plasma–mass spectrometry (Nu-ATTOM ES). NASS-5 (Seawater-certified reference material) and CASS-6 (Near Shore Seawater reference material) were analyzed to check the accuracy and reproducibility of the sample analysis. The analytical precision was monitored by repeated measurements of sample/standard and the calculated RSD was less than 3%.



The porewater sulfate concentration was measured from the $\Sigma HS^-$ free supernatant solution
using a Metrohm ion chromatograph (Basic IC plus 883) equipped with a suppressed
conductivity detector (Metrohm, IC detector 1.850.9010). A mixed solution of 1 mM $NaHCO_3$
and 3.2 mM $Na_2CO_3$ was used as the eluent, and 0.2 N $H_2SO_4$ was used as the suppressor
regeneration fluid. The samples were diluted 1000-fold with 18 MΩ ultrapure water prior to
analysis. The calibration line was prepared using a standard IC sulfate solution from a 100-
ppm mixed anion standard (Fluka), and the sample reproducibility was $\pm$ 0.1%. Quantification
of dissolved sulfide ($\Sigma HS^-$) was carried out following the methylene blue method (Cline et al.,
1969). Absorbance was measured at 670 nm on a spectrophotometer (Chemito Spectrascan
UV-2700). Sodium sulfide nonahydrate (≥99.99% purity, Merck) was used for the preparation
of calibration standards. The analytical error based on replicate standard measurements was <

159  3%.

**2.3 Solid phase analysis**


Iron extractions were conducted under anaerobic conditions using frozen sediment samples
following ascorbate and sodium dithionite steps. The leaching solutions were nitrogen-flushed
before extraction. The iron concentrations in ascorbate leached fraction ($Fe_{Asc}$) were
determined following previous studies (Hyacinthe et al., 2006; Raiswell et al., 2010) using
Atomic Absorption Spectrometer (Agilent-240AA) using air-acetylene flame. The sample
reproducibility for replicate analysis was within 5 %. The $Fe_{Asc}$ free sediment residue was
leached for 2 hrs using sodium dithionite buffer for determination of dithionite extractable iron
($Fe_D$) content following previous works (Mehra and Jackson, 1960; Canfield et al., 1989;
Volvaikar et al., 2020). The $Fe_D$ content was determined using ferrozine-complexometry
technique on a Chemito spectrophotometer (Spectroscan UV 2700). The calibration line was
prepared from pure ferrous ammonium sulfate (Merck) and absorbance was measured at 515
nm. The replicate for $Fe_D$ analysis yielded sample reproducibility within 6 %. The total



inorganic carbon content was measured using a UIC carbon coulometer (CM 5130). Ultrapure
$CaCO_3$ from Sigma-Aldrich was used as a standard reference material for measurement
(Carbon content: $12.0 \pm 0.25\%$). The measurement of total carbon (TC) content was carried out
on freeze-dried and desalinated samples using an EA1112 elemental analyzer (Thermo Fisher
Scientific, Germany). The total organic carbon (TOC) content (Figure S2; Table S2) was
calculated by subtracting TIC from TC. NC soil was used as a calibration standard for TC. The
reproducibility for TC in soil standards B2184 and B2152 was found to be $2.11 + 0.1$ % and
$1.53 + 0.07$ % respectively. The carbon and nitrogen isotope ratio of TOC ($\delta^{13}C_{TOC}$) and
nitrogen ($\delta^{15}N$) were measured on decarbonated samples using a Delta-V-plus isotope ratio
mass spectrometer coupled with an elemental analyzer (Thermo Flash EA 2000). The carbon
and nitrogen isotope ratios are reported as per mil (‰) deviations from the isotopic composition
of VPDB and atmospheric $N_2$ with reproducibility better than $\pm 0.3$‰.

## 3 Results and Discussion

### 3.1 Evidence for $SO_4^{2-}$ and Fe-Mn-AOM

Based on the porewater $SO_4^{2-}$ and Fe-Mn concentration profiles (Figures 2a-2e; Table S1), the
sediment core can be divided into three distinctive diagenetic regimes (zone-i, zone-ii, and
zone-iii). Zone-i (63 to 303 cmbsf) is characterized by $Fe^{2+}$ and $Mn^{2+}$ enrichment ($Fe^{2+}$: 10.75
to 361.3 μM and $Mn^{2+}$: 0.28 to 7.39 μM) at high $SO_4^{2-}$ concentrations (0.5 to 18.2 mM); zone-
ii (303 to 463 cmbsf) is characterized by $SO_4^{2-}$ depletion (0.4 to 1.45 μM) coupled with $Fe^{2+}$
and $Mn^{2+}$ enrichment ($Fe^{2+}$: 14.9 to 387.5 μM and $Mn^{2+}$: 0.41 to 10.17 μM); and zone-iii (463-
523 cmbsf) shows $SO_4^{2-}$ depletion (below detection limit to 0.8 μM) coupled with depleted $Fe^{2+}$
and $Mn^{2+}$ concentrations ($Fe^{2+}$: 11.85 to 58.2 μM and $Mn^{2+}$: 1.13 to 2.68 μM). The $\Sigma HS^-$
concentrations remain consistently high (163 to 10,385 μM) below 63 cm from the seabed.



The SMTZ is identified within an approximate depth window of 263 to 303 cmbsf (Figures 2a
and 2b). Within the SMTZ, AOM activity is identified by simultaneous depletion in $CH_4$
concentrations (188.9 to 503.2 μM) and C-isotope ratios of $CH_4$ ($\delta^{13}C_{CH4}$: -105.6 to -98.6 ‰)
and DIC ($\delta^{13}C_{DIC}$: -38 to 35.2 ‰). The AOM-driven depletion in $\delta^{13}C_{CH4}$ and $\delta^{13}C_{DIC}$ values
within the SMTZ may be attributed to (i) DIC back flux during AOM (Yoshinaga et al., 2014)
(ii) concurrent methanotrophic and methanogenic activity (Borowski et al., 1997), (iii) the
ability of ANMEs to perform as facultative methanogens (Pohlman et al., 2008), and (iv)
intracellular reaction reversibilities along enzymatic AOM pathway (Wegener et al., 2021).
Above SMTZ, the upward increasing trend in $\delta^{13}C_{CH4}$ values may be attributed to $^{13}C$
enrichment (Figure 2b; Alperin et al., 1988; Whiticar et al., 1996; Martens et al., 1999;
Whiticar, 1999) in the residual methane (kinetic isotope effect) due to methane oxidation and
lack of carbon recycling at high $SO_4^{2-}$ concentrations ($SO_4^{2-}$>0.5 mM; Yoshinaga et al., 2014;
Wegener et al., 2021). Below the SMTZ, the methanogenesis trend is superimposed by depths
of intermittent AOM activity evident from depletion in $CH_4$ concentrations (1.04 to 4.19 mM)
coupled with a sharp decrease in $\delta^{13}C_{CH4}$ (-17.95 to -61 ‰) and $\delta^{13}C_{DIC}$ values (-1.8 to - 4.1
‰) and increase in DIC concentrations (1 to 2.7 mM). The cross plot of DIC concentrations vs
$\delta^{13}C_{DIC}$ values for the overall core and the Fe-Mn-AOM specific points show negative
correlations (Figure 3). The intermittent increase in DIC concentrations and decrease in $\delta^{13}C_{DIC}$
values can also be identified within the sulfatic zone-i.



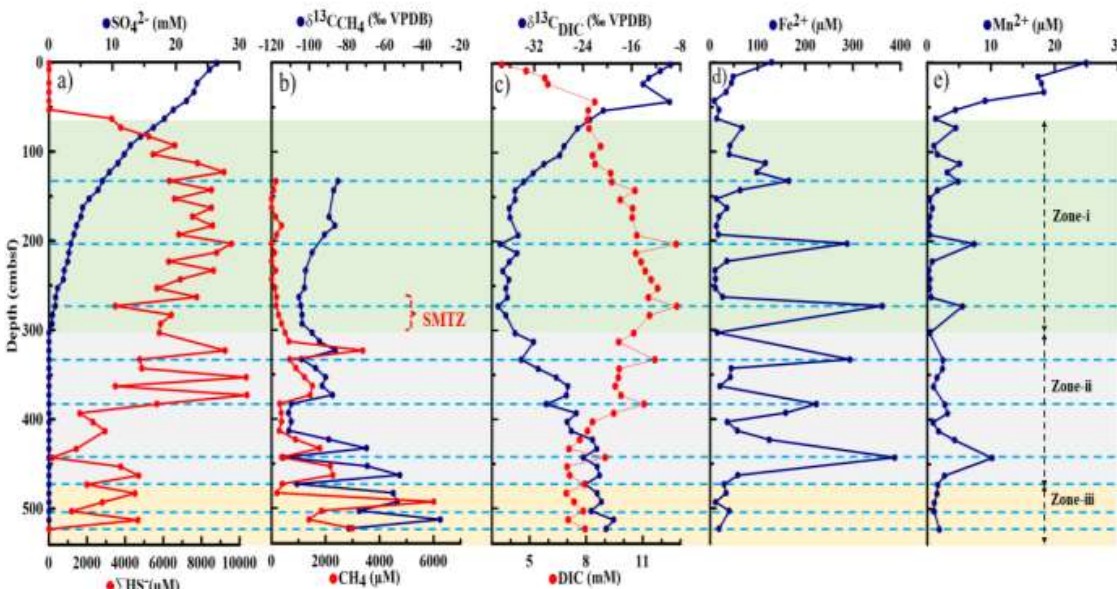


**Figure 2**. Figure showing Fe-Mn-AOM and $SO_4^{2-}$-AOM activities by means of pore fluid chemistry in seasonally hypoxic coastal sediment of the Eastern Arabian Sea. (a) Porewater $SO_4^{2-}$ (mM) and $\Sigma HS^-$ (µM) concentrations. (b) Dissolved $CH_4$ concentrations (µM) and $\delta^{13}C_{CH4}$ values. (c) Dissolved DIC concentrations (mM) and $\delta^{13}C_{DIC}$ values (d, e) Dissolved Fe and Mn (µM) concentrations. The blue dashed lines represent the depth layers with AOM activities. The red curly bracket depicts SMTZ. The different color shades represent zone-i, zone-ii, and zone-iii.



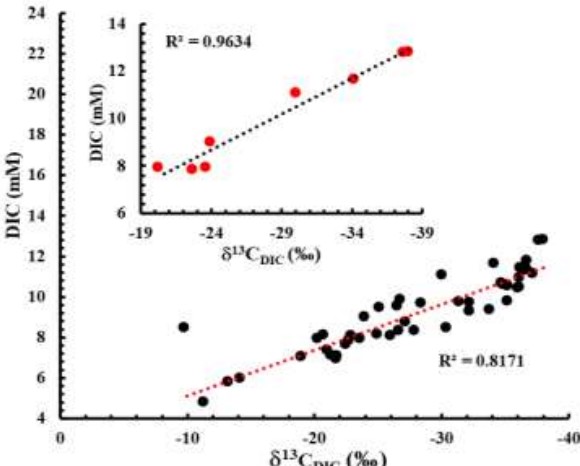

222

**Figure 3.** Cross plot of DIC concentration (mM) vs $\delta^{13}C_{DIC}$ (‰) values in the studied sediment core

(represented by black dots with an $R^2$ value of 0.82). The inset cross plot of DIC vs. $\delta^{13}C_{DIC}$ (represented

by red dots with an $R^2$ value of 0.96) corresponds to values specific to Fe-Mn-AOM (marked by blue

dashed lines in Figure 2).

The most intriguing observation of the present study is the intermittent increase in porewater
$Fe^{2+}$ and $Mn^{2+}$ concentrations (Figures 2d & 2e). Within zone-i and ii, the marked increase in
porewater $Fe^{2+}$ and $Mn^{2+}$ concentrations are associated with AOM activities. However, in zone-
iii, the AOM activities are not associated with significant porewater $Fe^{2+}$ and $Mn^{2+}$ enrichment.
The porewater $SO_4^{2-}$, $Fe^{2+}$, $Mn^{2+}$, $CH_4$, and DIC concentrations along with $\delta^{13}C_{CH4}$ and $\delta^{13}C_{DIC}$
depth profiles provide tell-tale evidence of both $SO_4^{2-}$-AOM and Fe-Mn-AOM in the present
study. Zonation of the geochemical profiles shows co-occurrence of $SO_4^{2-}$-AOM and Fe-Mn-
AOM in zone-i, and Fe-Mn-AOM in zone-ii and iii. The occurrence of Fe-Mn-AOM in $SO_4^{2-}$
rich zone has also been reported in an earlier study from the Dongsha area, South China Sea
(Li et al., 2019). These observations are further supported by an incubation study (Segarra et
al., 2013) where notable rates of Fe-Mn-AOM coincide with comparatively elevated levels of



$SO_4^{2-}$-AOM. Within zone-ii, where $SO_4^{2-}$ concentrations are reduced to 0.4 to 1.45 µM, the
$Fe^{2+}$ and $Mn^{2+}$ concentration spikes may be attributed primarily to Fe-Mn-(oxyhydr)oxides
fueling AOM activity (Riedinger et al., 2014; Egger et al., 2015; Egger et al., 2017).
In most of the previous studies (Crowe et al., 2009; Sivan et al., 2011; Slomp et al., 2013;
Riedinger et al., 2014; Egger et al., 2015; Egger et al., 2016a; Egger et al., 2016b; Egger et al.,
2017; Aromokeye et al., 2019; Vigderovich et al., 2019; Luo et al., 2020; Xiao et al., 2023),
AOM-driven $Fe^{2+}$-$Mn^{2+}$ enrichment has been reported at low to negligible $\Sigma HS^{-}$ concentrations
(Figure S1). However, the present study demonstrates tell-tale signatures of substantially high
porewater $Fe^{2+}$-$Mn^{2+}$ concentrations despite high $\Sigma HS^{-}$ concentrations. A similar observation
was also reported in previous studies (Ramírez-Pérez et al., 2015; Li et al., 2019). Equation-1
shows that for every mole of DIC produced via Fe-AOM, 8 moles of $Fe^{2+}$ are produced. The
DIC concentration perturbations (along the blue dashed lines) may be converted to equivalent
µM of $Fe^{2+}$ produced via $Fe^{3+}$ reduction in the pore waters. The stoichiometrically calculated
amount of $Fe^{2+}$ produced ranges from 11673 µM to 17163 µM in zone-i-ii and 6192 to 8259
µM in zone-iii. The measured porewater $Fe^{2+}$ concentrations are approximately 1.65-3% of the
calculated $Fe^{2+}$ in zone-i-ii and 0.2-0.6 % in zone-iii. Thus, the measured $Fe^{2+}$ concentration is
the residual $Fe^{2+}$ left after sulfidization (FeS/$FeS_2$: Hensen et al., 2003; Treude et al., 2014;
Peketi et al., 2015). Therefore, the lack of significant $Fe^{2+}$ and $Mn^{2+}$ enrichment in zone-iii may
indicate complete consumption of $Fe^{2+}$ and $Mn^{2+}$ via sulfidization.
Additional processes that may enhance the metal concentrations in the sediment pore waters is
the presence of metal (Fe-Mn) sulfide collides (Rickard et al., 2007) as nanoparticles.
Dissolution of porewater Fe-Mn sulfide nanoparticles (Morse et al., 1999; Olson et al., 2017)
in acid during sample preparation results in the simultaneous occurrence of $Fe^{2+}$ and $Mn^{2+}$
spikes in porewater. However, if the observed $Fe^{2+}$-$Mn^{2+}$ concentration spikes in this study are



merely an analytical artifact, then under the presence of such high dissolved sulfide
concentrations, $Fe^{2+}$-$Mn^{2+}$ spikes would be expected at all depths. The absence of Fe-Mn spikes
corroborated with distinct increase in DIC concentration and decrease in $\delta^{13}C_{DIC}$ values rules
out such artifact generation.

**3.2 Possible limiting factors and focused Fe-Mn-AOM**

An essential prerequisite for Fe-Mn-AOM is the presence of appreciable quantities of Fe-Mn-
(oxyhydroxides) to react with $CH_4$ (Beal et al., 2009) as evident from the stoichiometric ratios
(Eq. 1). Results from iron extractions indicate the presence of significant amount of iron in
highly reactive/bioavailable ($Fe_{Asc}$) (Figure 4b) and dithionite extractable phases ($Fe_D$) (Figure
4c, Table S2). The $Fe_{Asc}$ and $Fe_D$ contents in the sediment range from 0.01 to 4.27 mg/g (Avg:
1.2±0.86 mg/g) and 0.54 to 7.2 mg/g (Avg:3.03±1.36 mg/g) respectively. The porewater $Fe^{2+}$
concentration spikes are associated with marked depletion in $Fe_{Asc}$ content, indicating
consumption of ferrihydrite and other bioavailable iron through Fe-AOM (Figure 4). However,
the presence of substantial content of both $Fe_D$ and $Fe_{Asc}$ throughout the core length suggests
Fe-AOM is not limited by reactive iron content. In the present study, the high reactive iron flux
in the shelf zone may be attributed to rapid sedimentation rates (150 to 1500 cm/ky) and fluvial
input from Fe-Mn-rich provenances (Fernandes et al., 2020). Additionally, the presence of $CH_4$
throughout the Fe-Mn-AOM zones suggests that neither $CH_4$ nor reactive Fe. are limiting
factors responsible for the focused occurrences of Fe-AOM in the sediment core.



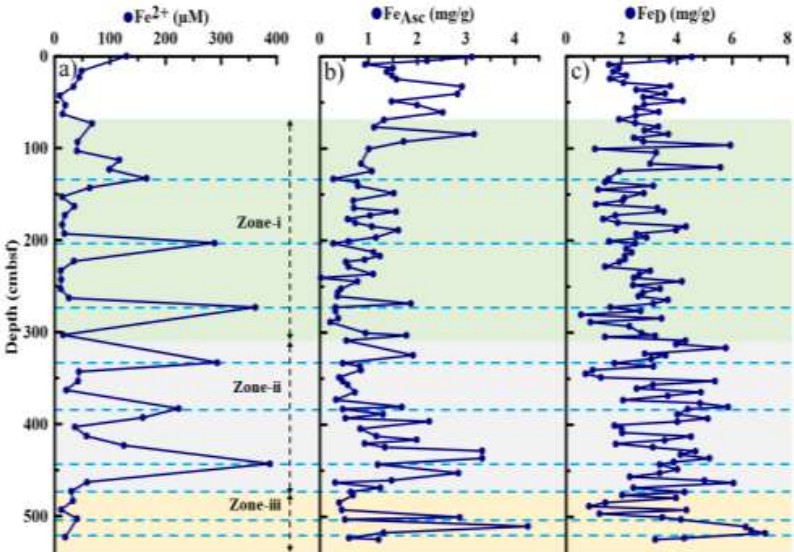

**Figure 4.** Depth profiles of (a) Porewater $Fe^{2+}$ concentrations (µM). (b) solid phase ascorbic acid leachable iron ($Fe_{Asc}$) and (c) sodium dithionite leachable iron ($Fe_D$) contents (in mg/g). The blue dashed lines represent the depth layers with AOM activities. The different color shades represent zone-i, zone-ii, and zone-iii.

One of the plausible factors responsible for the occurrence of focused Fe-Mn-AOM in distinct sediment layers could be the spatiotemporal enrichment of the microbial population driving Fe-Mn-AOM. The latter in turn could be the result of the interplay between several biogeochemical factors. For instance, the shallow shelf off WCI is highly dynamic, experiencing drastic changes in water column redox conditions, marine productivity, fluvial fluxes of organic matter, sediment load, and extensive denitrification (Naqvi et al., 2000; Schott et al., 2001; Naqvi et al., 2006; Maya et al., 2011; Mazumdar et al., 2012; Fernandes et al., 2020). A large range in TOC contents (1.45 to 31.3 mg/g), (TOC/TN)$_{molar}$ ratios (3.47 to 27.32), $\delta^{13}C_{TOC}$ (-20.69 to -25.93 ‰), and $\delta^{15}N$ values (-3.6 to 7.7 ‰) in the present and previous studies (Mazumdar et al., 2012; Fernandes et al., 2020) indicate marked temporal variation in the fluvial and marine organic matter fluxes and denitrification conditions in WCI (Figure S2,



Table S2). Previous studies (near SSD070/7/GC6; Figure 1b) investigating carbon and nitrogen
stable isotopes of suspended particulate organic matter (SPOM) in the estuary (Bardhan et al.,
2014) and shelf zone (Maya et al., 2011) of WCI revealed significant intra-annual variations in
$\delta^{15}N$ (estuary: 0.69 to 7.26 ‰; shelf: -4.17 to 10.43 ‰) and $\delta^{13}C$ (estuary: -30.14 to -19.52 ‰,
shelf: -17.64 to -26.74 ‰) throughout the year. These variations reflect the complex and
dynamic nature of biogeochemical processes and organic matter sources in the coastal waters
of the WCI. Corroboratively, significant variations in the diversity, abundance, and activity of
microorganisms, attributable to seasonal differences in nutrient availability, have been recorded
in the water column between monsoon and non-monsoon seasons (Gomes et al., 2019; Naik et
al., 2024; Parab et al., *bioRxiv*). Spatiotemporally contrasting biogeochemical conditions of
shallow coastal waters have profound influence on the structure and function of underlying
sedimentary microbiomes (Bhattacharya et al., 2021). An earlier study (Orsi et al., 2017) from
the North Eastern Arabian Sea oxygen minimum zone reported stratigraphic microbial
distribution attributed to palaeoceanographic conditions. It was also proposed that focused
abundances of microbial communities in sediment layers are expected to be most pronounced
in the coastal region due to drastic variations in marine redox conditions and terrestrial fluxes.
In the context of the biogeochemical phenomenon revealed here, we hypothesize a dominant
role of the localized abundance of metal-reducing bacterial/archaeal communities in restricting
Fe-Mn-AOM activities into specific sedimentary strata. The focusing of microbial activity in
different sediment layers may be attributed to factors such as past environmental conditions
and depositional processes (Parkes et al., 2000; Orsi et al., 2017; Hoshino et al., 2020). Seasonal
coastal hypoxia coupled with a wide range of organic matter (marine to terrestrial) fluxes may
play   important   roles   in   stratigraphic   microbial   distributions   and   heterogeneity   in
biogeochemical processes.





321 Our hypothesis is supported by metagenomic sequencing (Sarkar et al., *in prep*; Table S3)

322 carried out in a ~3 m long, sulfide-rich, sediment core (SSK42/9; water depth: 31 m; Figure

323 1b), collected from a nearby shallow coastal site (Bhattacharya et al., 2021) which shows the

324 presence of two species-level entities affiliated to the uncultivated archaeon

325 *Candidatus* Methanoperedens (GenBank assembly numbers GCA_003104905.1 and

326 GCA_001317315.1) reported thus far for potential abilities to harness anaerobic methane

327 oxidation to the reduction of Fe (III) (Ettwig et al., 2016; Cai et al., 2018). However, the final

328 microbiological corroboration of the focused Fe-Mn-AOM phenomenon hinges on the future

329 enrichment of the genome sequence database for microorganisms having definite and complete

330 Fe-Mn-AOM attributes.

331 Being the world's largest coastal hypoxic zone covering an area of 0.18 million sq. km, the net

332 influence of Fe-Mn-AOM could be significant in sedimentary methane consumption. A

333 schematic representation of $CH_4$-S-Fe-Mn cycling in coastal hypoxic sediments of the Eastern

334 Arabian Sea is presented in Figures 5a, 5b and 5c respectively.



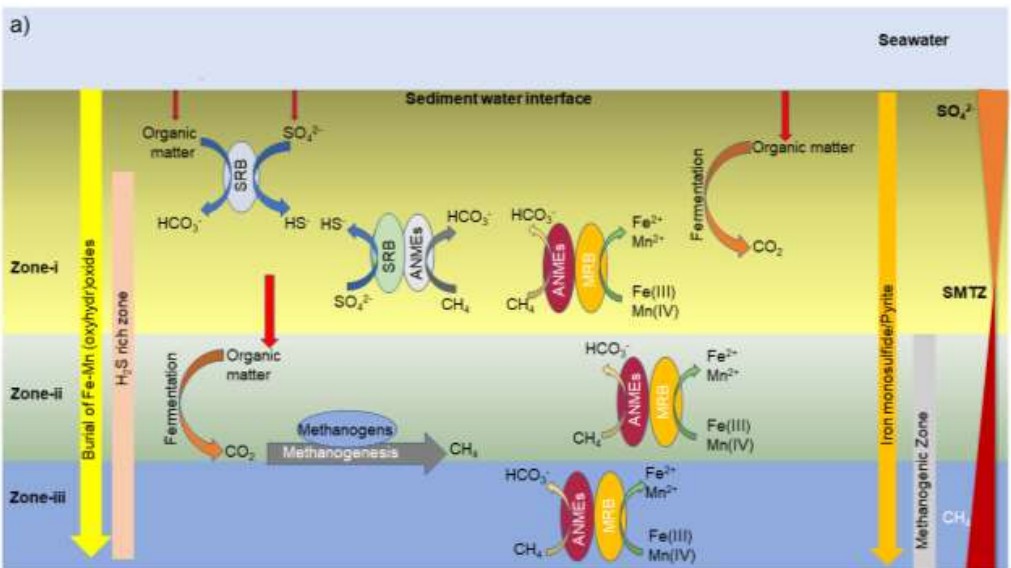

SRB: Sulfate reducing bacteria; MRB: Metal reducing bacteria; ANME: Anaerobic methanotrophic archaea

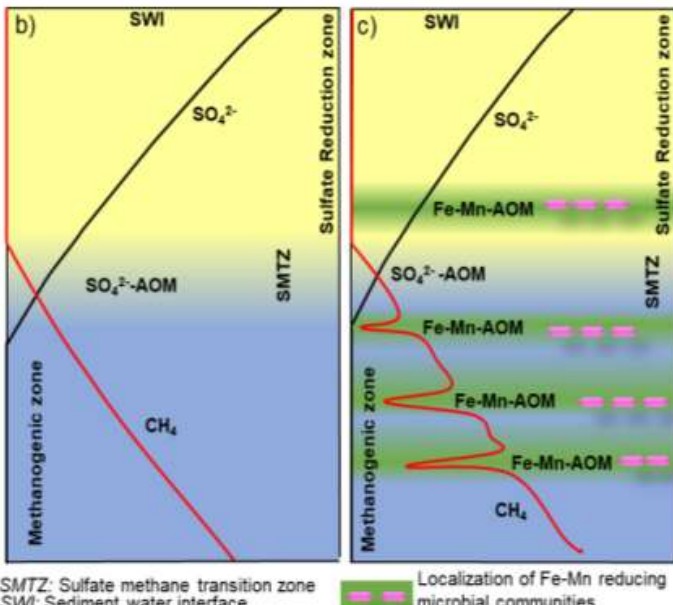

*SMTZ: Sulfate methane transition zone*  |  Localization of Fe-Mn reducing
*SWI: Sediment water interface*  |  microbial communities


**Figure 5**. (a) Schematic representation of Fe-Mn-AOM and $SO_4^{2-}$-AOM activities within the sediment
column. The figure depicts different zonation within the sediment column, showing the co-occurrence
of $SO_4^{2-}$-AOM and Fe-Mn-AOM in zone-i and Fe-Mn-AOM in zone-ii-iii. Schematic representation of





pore fluid vertical profiles in the absence (b) and presence (c) of Fe-Mn-AOM activity. The green-
shaded regions in the methanogenic zone represent potential Fe-Mn-AOM depth layers, controlled by
the localization of metal-reducing microbial communities.

## 4 Conclusions

The comprehensive geochemical investigation of the sediment core (5.25 m) retrieved from the
seasonally hypoxic shelf sediment off the eastern Arabian Sea (west coast of India) has
unveiled compelling evidence of focused Fe-Mn-AOM occurring at multiple depths beneath
the seafloor. The accumulation of porewater $Fe^{2+}$ and $Mn^{2+}$ despite elevated concentrations of
$\Sigma HS^{-}$, suggests the predominant production of $Fe^{2+}$ and $Mn^{2+}$ rather than consumption through
metal sulfidization processes. This study demonstrates that Fe-Mn-AOM remains a significant
biogeochemical phenomenon even within $SO_4^{2-}$ and $\Sigma HS^{-}$-rich sediments. Our findings are also
relevant in assessing the consumption of $CH_4$, a greenhouse gas in organic-rich coastal
sediments, thus having implications for climate change research. This study presents a new
perspective by documenting the biogeochemical heterogeneity in the occurrence of Fe-Mn-
AOM, possibly attributed to deposition-controlled Fe-Mn reducing microbial population
distribution in a highly dynamic coastal environment sensitive to climate change. The findings
of the present study may have a far-reaching influence on coupled $CH_4$-Mn-Fe-S cycling in
expanding hypoxic coastal regions of the global ocean. Our findings lead to a hypothesis that
necessitates future examination of microbial communities in seasonally hypoxic sediments and
SPOM at high depth and temporal resolution which may uncover the complex interactions
between microbial life and their environment.
**Appendices:** No appendix is there in this manuscript.
**Code Availability:** No code was used in this manuscript.
**Data availability:** All the data used in this study is incorporated in the supplementary text as
table S1, table S2 and table S3.




**Sample availability:** Samples are available with the author.

**Video supplement:** No video component is there in this manuscript.

**Author Contributions:** K.S., A.P., and A.M. designed the research; A.M. obtained the funding; K.S., A.Z., and S.P.K.P. analyzed the data; A.G., M.S., and J.M. carried out sampling; and K.S., A.P., and A.M. wrote the paper.

**Competing interests:** The authors declare that they have no conflict of interest.

**Acknowledgments**

The authors thank the Director of CSIR-NIO for supporting this study. We sincerely thank the Ministry of Earth Sciences (GAP2303) for funding the program and the Council of Scientific and Industrial Research (CSIR) for providing a research fellowship to K.S. We are grateful to Dr. V.V.S.S. Sarma for helping with carbon and nitrogen isotopic measurements. We also thank Dr. Wriddhiman Ghosh for his valuable suggestions on the manuscript. Sincere thanks to Mr. Sidharth Vernekar, Mr. Harish Kumar, and crewmembers for onboard support. Thanks are due to Mr. Robin John and Mrs. Teja Naik for helping with HR-ICPMS and carbon coulometry analysis.

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

**Supplementary Data**. For figures (Figures S1 and S2), tables (Tables S1, S2 and S3), and
supporting text, please see supplementary data file.
