# Peer review of "Enigmatic Fe-Mn-fueled Anaerobic Oxidation of Methane in sulfidic coastal"

_EGUsphere, 2024_

## Author Comment (AC1)

**Response to comments of Reviewer 1**

**Review Sivan et al.**

**General comments**

Sivan et al. investigate biogeochemical processes, in particular the anaerobic oxidation of methane coupled with Fe and/or Mn reduction (Fe-Mn-AOM), in seasonally hypoxic coastal sediments of the Eastern Arabian Sea. The main finding is that Fe-Mn-AOM is observed throughout the sediment core at several specific depths above and below the sulfate-methane transition zone (SMTZ). It is the first study to report Fe-Mn-AOM under sulfidic conditions. The authors argue that the activity of Fe-Mn-AOM at very specific depths is controlled by the distribution of metal-reducing and methanotrophic microbial communities in the sediment core.

While reading the manuscript, several concerns and questions arose. These could be addressed with major revisions. Overall, the article is of good quality, but the written language and some of the figures could be improved.

**Comments 1:** First of all, the present study lacks a clear/specific research question or hypothesis. In lines 82-84, the authors only describe which aspects (driving factors fueling Fe-Mn-AOM) were investigated. However, the research gap and the specific research question of the study are not explained at this point.

**Reply:** Thank you very much for the comment. Your feedback is greatly appreciated. The present study aims to investigate the occurrence and behavior of Fe-Mn-driven anaerobic oxidation of methane (AOM) in seasonally hypoxic coastal sediments of the Western Continental Shelf of India (WCSI; Eastern Arabian Sea), particularly under non-sulfatic and mild to moderate sulfatic conditions. The WCSI experiences distinct seasonal variations, including hypoxia and denitrification over the inner and middle shelf from June to October, driven by upwelling during the southwest monsoon (Naqvi et al., 2000; Schott et al., 2001). For the rest of the year, the region is characterized by oligotrophic conditions and normoxia above sediment-water interface. During the southwest monsoon months, the WCSI receives significant inputs of both terrestrial and marine organic matter due to enhanced land runoff and marine productivity associated with upwelling (June to September). Consequently, the WCSI displays contrasting seasonal biogeochemical conditions including variations in hydrography and depositional features (Naqvi et al., 2006; Mazumdar et al., 2012).

The silt-clay-rich sediments of the study area is characterized by TOC content limited within 1.45 to 31.3 mg/g. Both $(TOC/TN)_{molar}$ ratios (3.47 to 27.32) and $\delta^{13}C_{TOC}$ values (-20.69 to -25.93 ‰) from the present and previous studies (Mazumdar et al., 2012; Fernandes et al., 2020) indicate marked temporal variation in the fluvial and marine organic matter fluxes in WCSI. Moreover, the previous (Mazumdar et al., 2009; Fernandes et al., 2020) and present observations have shown remarkable fluctuations and covariance in methane concentration and $\delta^{13}C_{CH4}$ below the SMTZ in WCSI. This observation could not be explained by variation in the content and nature of organic matter, lithology (data presented in figure 2), or diffusional loss of methane from the sediment core. The research target of the present work is primarily to explain the reason for these fluctuations which has been observed both in the non-sulfatic and moderately sulfatic conditions. So, in order to explain this process, porewater $Fe^{2+}$, $Mn^{2+}$, DIC concentrations, $\delta^{13}C_{DIC}$ values, and several other solid phase parameters (Fe speciation, Mn bulk concentrations, $FeS_2$ concentrations) were investigated. In addition, despite studies on Fe-Mn-AOM in coastal sediments, the feasibility and nature of Fe-Mn-AOM in seasonally hypoxic coastal environments remain underexplored. Moreover, how Fe-Mn-AOM is influenced by past contrasting biogeochemical conditions remains understudied. This study aims to understand this gap by providing new insights into the nature and feasibility of Fe-Mn-AOM in dynamic depositional and redox environments with significant concentrations of both sulfide and sulfate.

Comment 2: The authors state that Fe-Mn-AOM is thermodynamically more favorable than $SO_4^{2-}$-AOM (lines 52-53). Dissimilatory Fe reduction (DIR) however is thermodynamically more favorable compared to Fe-AOM. How do the authors rule out that the Fe peaks (and Mn peaks) above the SMTZ (where $CH_4$ concentrations are close to zero) are not due to DIR (or Mn reduction)? The solid-phase data show that high levels of TOC and reactive Fe oxides are present. Furthermore, Mn-AOM is thermodynamically more favorable compared to Fe-AOM (Eq. 1 and however the authors do not distinguish between Fe-AOM and Mn-AOM. Therefore, does it make sense that the Mn and Fe peaks are found at exactly the same depth (especially if the presence of Fe oxides is not the limiting factor)?

**Reply:** Thanks for the suggestion. There are several subquestions within this. We have answered one by one.

A) Dissimilatory iron reduction (DIR)

Dissimilatory iron reduction (DIR) is assumed to be an important source of dissolved iron (eq.1) in anoxic marine sediments (Lovley & Phillips, 1986; Roden & Wetzel, 2003).

$$2Fe^{3+} + \text{organic matter/}H_2\text{/humic acids} \rightarrow 2Fe^{2+} + HCO_3^- /CO_2/2H^+ \qquad \text{(eq.1)}$$

In support of our argument, kindly see Figure 1 given below. The figure 1 shows green shaded zones of porewater $Fe^{2+}$, $Mn^{2+}$ enrichment coupled with TOC content, $\delta^{13}C_{TOC,}$ and $\delta^{13}C_{DIC}$ values above the SMTZ (marked by pink shaded region). The Fe-Mn-rich zones do not show any unequivocal correlation with TOC content compared to the TOC content in the metal-depleted depths. In addition, the $\delta^{13}C_{DIC}$ within the Fe-Mn-AOM region ranges from -32 to -37 ‰ which is 11 to 15 ‰ more depleted than what is expected via a purely Organoclastic Fe-Mn-reduction pathway. In a purely DIR pathway, the $\delta^{13}C_{DIC}$ value is expected to be slightly enriched or close to bulk $\delta^{13}C_{TOC}$ values (Malinverno and Pohlman, 2011; Fernandes et al., 2018). However, we agree with the reviewer that dissimilatory Fe-Mn reduction may also be taking place above the SMTZ. This process may be happening along with the Fe-Mn-AOM process as suggested by depleted $\delta^{13}C_{DIC}$ values. In agreement with the reviewer, we shall be incorporating the possibility of Dissimilatory Fe and Mn reduction in the revised text.

B) The Fe-Mn covariance in the porewater indicates that both Fe and Mn are responsible for AOM. Distinguishing and quantifying the extent of Fe and Mn in AOM activity have not been attempted in the manuscript because we feel microbiological data may be required to attempt which is currently beyond the scope of the paper. Future expeditions in this region will be in collaboration with microbiologists. In the present work, we have carried out speciation of Fe, however we have not been successful with Mn solid phase speciation. We would sincerely request the reviewer to bear with this shortcoming. Suitable revisions will be made to the text.

[Figure]

*Figure 1: Depth profiles of A) TOC (mg/g), porewater (B) Fe²⁺, (C) Mn²⁺, (D) DIC concentration and δ¹³C_DIC, (E) δ¹³C_TOC and (F) TOC/TN_(molar) from the present study. The green shaded zones represent porewater Fe²⁺ and Mn²⁺ enrichment coupled with TOC content, δ¹³C_TOC, and δ¹³C_DIC values above the SMTZ (marked by pink shaded region).*

Comment 3: My main concern is that no information about the sediment composition/properties (e.g., lithology, grain size, bulk element content, porosity) is given. Only the sedimentation rate is reported (line 227). However, the given range (and hence the resulting sediment age) is also an order of magnitude different. Information on sediment composition is particularly important in an area with changing depositional conditions (as also described by the authors in lines 310-320). How did the depositional conditions change in the past?

**Reply:** The sedimentation took place in the hypoxic inner shelf off the Eastern Arabian Sea. Based on the previous ²¹⁰Pb dating from the present coring site, the sedimentation rate ranges from 0.19 cm/yr (below 40 cmbsf) to 1.5 cm/yr (above 40 cmbsf) (Sebastian et al., 2017). The grain size analysis data shows a dominantly clayey silt sediment type where the clay minerals are composed of kaolinite and montmorillonite with very low illite content. The median grain size (D50) data shows a small variability of 6.66 to 17.43 um (Figure 2D). The zones are homogenous in terms of porosity except for some spikes in between. However, those spikes do not correlate with any of the porewater Fe²⁺-Mn²⁺ spikes. Moreover, the Fe²⁺/Mn²⁺ enrichment zones do not show any correlation with TOC content and TOC/TN_(molar) ratios. The

sedimentological characteristics in the core don't show any obvious correlation with the biogeochemical observations.

[Figure]

*Figure 2: Depth profiles of (A) porosity, porewater (B) $Fe^{2+}$ (C) $Mn^{2+}$ (D) median grain size (D50) (E) TOC (mg/g) (F) TOC/TN(molar) in the present study (SSD070/7/GC6).*

Comment 4: Looking at the Table S1 in the Supplemental Material, it is noticeable that the pore-water sample for the trace metal measurement above the Fe and Mn peaks is often missing. Is there a reason for this? Was less pore water extracted at the depths with the Fe and Mn peaks? Could this be related to differences in sediment composition? It is also striking that the Fe and Mn peaks are so regular.

In addition, Zones i to iii were defined only based on the presence or absence of sulfate and the Fe and Mn peaks. For example, are the zones homogeneous in terms of sediment composition?

**Reply:** The volume of porewater extracted in this core by centrifugation at 8000 rpm was between 6 to 7 ml. This volume was used for all chemical analysis including gravimetry. For the two samples mentioned by the reviewer, the porewater volume was not enough for metal concentration measurements. We have furnished here sediment porosity and sediment grain size data along with the porewater $Fe^{2+}$-$Mn^{2+}$ data (Figure 2). The sediments of this region are clayey silt with median grain size (D50) showing a small variability of 6.66 to 17.43 um (Figure 2D). The clay mineralogical studies show a uniform mineralogy composed of kaolinite and montmorillonite. The zones are homogenous in terms of porosity except for some spikes in between. Both porosity and grain size data do not show any obvious correlation with porewater $Fe^{2+}$-$Mn^{2+}$ data. In the previous figure (Figure 2), we have shown that the TOC data also doesn't show any unequivocal correlation with porewater $Fe^{2+}$-$Mn^{2+}$ data.

The sedimentological characteristics in the core don't show any obvious correlation with the biogeochemical observations.

Comment 5: Is the calculation in lines 247-256 based on the assumption that DIC release is due to Fe-AOM only? There are certainly other degradation processes of the organic material that lead to the release of DIC. Therefore, the value of 11000 μM is probably significantly overestimated. If so much Fe is supposed to have reacted with HS⁻, there should be a high enrichment of Fe sulfides in the corresponding layers. Was the Fe monosulfide, pyrite, or total sulfur content also determined? The solid-phase contents might be essential to support the hypothesis.

**Reply:** Thank you so much for pointing out the possible iron sulfide enrichment that can occur in Fe-Mn-AOM zones. (A) Fe monosulfide (FeS)

Fe monosulfide could not be detected in this core in spite of using 6N room temperature HCl extraction (Canfield et al., 1986). Previous studies on sulfide speciation from this region also did not report (Fernandes et al., 2020) iron monosulfides which may be attributed to complete pyritization due to high hydrogen sulfide precipitation. In another paper (Sivan et al. in *prep*) the sulfidization pathway in this region will be discussed.

(B) The iron sulfide content (CRS; chromium reducible sulfur)

Regarding your concern about the potential overestimation of Fe-AOM's contribution to DIC, we understand that other processes could contribute to the DIC release, and thus the value may be maximum Fe production via Fe-AOM. However, the observed exact correspondence of DIC concentration and AOM zones (depleted $\delta^{13}C_{DIC}$ and $\delta^{13}C_{CH4}$ values) suggests that DIC production is primarily controlled by AOM. Since these AOM zones are also accompanied by porewater Fe-Mn increase, we attribute the DIC enrichment primarily to Fe-Mn-AOM (Luo et al., 2020). Furthermore, the calculated ferrous productions are 17163.18, 11673.09, 16743.20, 12883.82, 16854.54, 8159.55, 6192.97, 7104.52 μM which is equivalent to CRS contribution of 2.06, 1.4, 2.01, 1.54, 2.02, 0.97, 0.74, 0.85 mg/g. The measured CRS content in those Fe-Mn-AOM peak depths corresponds to 19.66, 47.3, 20.33, 51.3, 57.7, 72.88, 50.98, 73.8 mg/g which is significantly high compared to CRS which might have formed from Fe produced via Fe-AOM. Moreover, the bulk CRS content throughout the sediment core ranges from 3.32 to 102.3 mg/g (mean $35\pm 17$ mg/g) (Figure 3D). It's important to note that pyritization in sediments is a cumulative process throughout sediment diagenesis, primarily controlled by the rate of microbial sulfate reduction, sedimentation rate, labile organic flux (Berner, 1985;

Raiswell and Berner, 1985; Wilkin and Barnes, 1997; Werne et al., 2003; Markovic et al., 2015), bottom water oxygenation which supports benthic fauna causing bioturbation and subsequent reoxidation of Fe-sulfide minerals (Chambers et al., 2000; Antler et al., 2019), and the availability of Fe that can react with sulfide (Jørgensen, 1982; Yücel et al., 2010; Zhu et al., 2016; Jørgensen et al., 2019). Additionally, in marine sediments, pyrite formation is influenced by the spatial availability of Fe and S redox reactions (Rickard and Luther, 2007). Both $HS^-$ concentration and isotope profile of $HS^-$ does not match with that of CRS in the present (Figure 3C, E) and previous studies (For eg: Raven et al., 2016; Fernandes et al., 2020) which itself reflects the cumulative effect of sulfidization on speciation and isotope ratios during sediment burial. However, porewater $HS^-$ concentrations and isotope ratios at any depth do not represent a cumulative record and tend to become enriched in $^{34}S$ with burial (Mazumdar et al., 2012). We do agree with the reviewer that some amount of Fe produced via Fe-AOM is definitely precipitating as iron sulfide. However, given the large range in $FeS_2$ content, partitioning of $H_2S$ in iron sulfide and organic bound sulfur (OBS) and the complexity involved in the cumulative nature of CRS and OBS makes quantification of Fe-AOM contribution towards additional FeS/CRS precipitation a little challenging especially through gravimetry. **However, we will include this aspect suggested by the reviewer in the revised text.**

[Figure]

*Figure 3: Depth profiles of porewater (A) $SO_4^{2-}$, $\Sigma HS^-$, (B) $Fe^{2+}$, (C) $\delta^{34}S_{SO4^{2-}}$, $\delta^{34}S_{\Sigma HS^-}$, (D) CRS and (E) $\delta^{34}S_{CRS}$ in SSD070/7/GC6 (present study).*

 If I understand correctly, the HS⁻ concentrations in core SSK42/9 (Bhattacharya et al., 2021) are significantly lower than in this study. Are Fe and Mn concentrations also available for this core? Do the profiles also show the Fe and Mn peaks? Are the samples listed in Table S3 from layers with high Fe concentrations? Otherwise, the comparison with the microbial data from core SSK42/9 does not necessarily support the hypothesis that the Fe-Mn AOM activities in the specific zones are solely due to the presence of specific microbial communities.

**Reply:** We agree that the HS⁻ concentrations in core SSK42/9 (Fernandes et al., 2020; Bhattacharya et al., 2021) are significantly lower, ranging from 41.2 to 1196 µM. Unfortunately, porewater Fe and Mn concentrations were not measured in SSK42/9. As a result, we cannot confirm whether the samples listed in Table S3 correspond to Fe and Mn peaks. **We have cited Bhattacharya et al. (2021) just to show that Fe-Mn reducing microbial communities are present in these shelf sediments.** Whether these bacteria are concentrated in specific zones can also be proved in microbiological studies in future work. We agree with the reviewer that in the absence of porewater $Fe^{2+}$-$Mn^{2+}$ data, Bhattacharya et al. (2021) has limitations. We will not be elaborating on the data but instead, limit the discussion to show evidence of the presence of Fe-Mn-reducing microbial communities in these shelf sediments.

Comment 7: The figures in the manuscript are very pixelated and some figures could be improved (please see specific comments).

**Reply:** Thank you for your feedback regarding the quality of the figures in the manuscript. We apologize for the pixelation and any lack of clarity. During manuscript submission, we uploaded high resolution figures. However, the merged manuscript probably becomes a low resolution. **We will replace the current figures with higher-resolution versions in the revised manuscript.**

**Specific comments**

Although not required by the journal, it would be useful to separate the Results and Discussion sections for an easier understanding.

**Reply:** Thank you for your suggestion to separate the Results and Discussion sections. We will carefully consider this revision and reorganize the manuscript to present the Results and Discussion in separate sections to improve the overall understanding for the readers.

**Abstract**

Line 22: $\delta^{13}C_{CH4}$ and $\delta^{13}C_{DIC}$ were not introduced.

**Reply:** Thank you for pointing out that $\delta^{13}C_{CH_4}$ and $\delta^{13}C_{DIC}$ were not introduced before line 22. The text will be revised to include both.

**Introduction**

Line 61: Aromokeye et al. was published in 2020.

**Reply:** Thank you for the observation. Aromokeye et al. (2019) will be changed to Aromokeye et al. (2020) in line 61 in the revised manuscript.

Line 70: The sentence "The global distribution of Fe-Mn-AOM is plotted in Figure 1a" is a bit lost here. The global occurrence of Fe-AOM should be better integrated and discussed in the introduction.

**Reply:** The text will be revised accordingly.

Line 82-84: The specific research questions are missing. In addition, this is almost the same sentence as lines 70-71.

**Reply:** Thank you very much for the comment. Your feedback is greatly appreciated. The present study aims to investigate the occurrence and behavior of Fe-Mn-driven anaerobic oxidation of methane (AOM) in seasonally hypoxic coastal sediments of the Western Continental Shelf of India (WCSI; Eastern Arabian Sea), particularly under non-sulfatic and mild to moderate sulfatic conditions. The WCSI experiences distinct seasonal variations, including hypoxia and denitrification over the inner and middle shelf from June to October, driven by upwelling during the southwest monsoon (Naqvi et al., 2000). For the rest of the year, the region is characterized by oligotrophic conditions and normoxia above sediment-water interface. During the southwest monsoon months, the WCSI receives significant inputs of both terrestrial and marine organic matter due to enhanced land runoff and marine productivity associated with upwelling (June to September). Consequently, the WCSI displays contrasting seasonal biogeochemical conditions, including variations in hydrography and depositional features (Naqvi et al., 2006; Mazumdar et al., 2012).

The silt- clay rich sediments of the study area is characterized by TOC content limited within 1.45 to 31.3 mg/g. Both (TOC/TN)$_{molar}$ ratios (3.47 to 27.32) and $\delta^{13}C_{TOC}$ values (-20.69 to -25.93 ‰) from the present and previous studies (Mazumdar et al., 2012; Fernandes et al., 2020)

indicate marked temporal variation in the fluvial and marine organic matter fluxes in WCSI. The previous and present observations have shown remarkable fluctuations and covariance in methane concentration and $\delta^{13}C_{CH4}$ below the SMTZ. This observation could not be explained by variation in the content and nature of organic matter, lithology (data presented in the revised study) or diffusional loss of methane from the sediment core. The research target of the present work is primarily to explain the reason for these fluctuations which has been observed both in the non sulfatic and moderately sulfatic conditions. So, to explain this process, porewater Fe, Mn, DIC concentrations, $\delta^{13}C_{DIC,}$ and several other solid phase parameters (Fe speciation, Mn bulk concentrations, $FeS_2$ concentrations) were investigated. In addition, despite studies on Fe-Mn-AOM in coastal sediments, the feasibility and nature of Fe-Mn-AOM in seasonally hypoxic coastal environments remain underexplored. Moreover, how Fe-Mn-AOM is influenced by past contrasting biogeochemical conditions remains understudied. This study aims to understand this gap by providing new insights into the nature and feasibility of Fe-Mn-AOM in dynamic depositional and redox environments with significant concentrations of both sulfide and sulfate.

**Methods**

Line 108: For headspace methane analysis, the sediment was extracted using 50 ml cut syringes at an interval of 10 cm and transferred into 20 ml headspace vials filled with 3 ml of KOH and 3 ml of $NaN_3$ to trap $CO_2$ and [stop instead of arrest] microbial activities respectively.

**Reply:** Thank you for pointing out the wording issue in Line 108. **The text will be revised accordingly.**

Line 115: Which constituents exactly?

**Reply:** The porewater was stored at 4°C for measurement of $SO_4^{2-}$ concentration, sulfur isotope ratio of $SO_4^{2-}$ and $H_2S$, DIC, and stable carbon isotope composition of the DIC ($\delta^{13}C_{DIC}$).

Line 117: In what ratio were the samples acidified? How long after sampling were the trace metal aliquots acidified? Were the trace metal samples all diluted equally (i.e., 1:40)?

**Reply:** Immediately after sediment core recovery, sediment samples from the cores were centrifuged on board and different aliquots were collected under Argon atmosphere in a box (Figure 4) with a clear PVC strip curtain to extract porewater for chemical analysis. For trace metal analysis, porewater samples were acidified with 100 μl of concentrated supra-pure

HNO₃. All glass vials were flushed with argon, sealed, and stored at 4°C until shore-based analysis. All samples were diluted equally at 40 times for trace metal analysis.

[Figure]

*Figure 4: Schematic diagram showing the sampling protocol followed for porewater extraction followed in the present study*

Line 124: GC was not introduced.

**Reply:** Thank you for your observation. The suggested change will be made in the revised manuscript.

Line 161-162: Was the sediment thawed and homogenized before extraction? How much sediment and how much extraction solution were used? Was the water content taken into account when weighing the sediment?

**Reply:** Yes, the sediment for $Fe_{ASc}$ and $Fe_D$ extraction was thawed and homogenized before extraction following Ferdelman (1988) and Raiswell et al. (2010). All sediments were thawed at room temperature under a nitrogen shower box and extracted for Fe speciation. Sequential extractions were carried out under anoxic conditions. The leaching solutions were nitrogen-flushed before extraction. Around 0.5 to 1 g of sediment sample was treated with 50 g L⁻¹ sodium citrate (0.17 M), 50 g L⁻¹ sodium bicarbonate (0.6 M), and 10 g L⁻¹ of ascorbic acid (0.057 M). The $Fe_{Asc}$-free sediment residue was leached for 2 hrs using sodium dithionite buffer (50 g L⁻¹ buffered to pH 4.8 with 0.35 M acetic acid/0.2 M sodium citrate) for determination

of dithionite extractable iron ($Fe_D$) content following previous works (Mehra and Jackson, 1960; Canfield et al., 1989). The water content in the samples was determined separately and the reactive Fe content was determined on a dry weight basis.

**Results and Discussion**

Line 211-213: What does the negative correlation mean?

**Reply:** Thank you for the comment. Normally in any porewater profile, below SMTZ, a decrease in DIC content and an increase in DIC carbon isotope ratio is observed which may be attributed to hydrogenotrophic methanogenesis (Whiticar et al., 1999) that results in a negative correlation. On the other end, the AOM involves DIC production and depletion in the DIC isotope ratio which also results in a negative correlation (Yoshinaga et al., 2014; Egger et al., 2017; Luo et al., 2020). If required, this may be incorporated into the result/discussion.

Line 232-245: "Tell-tale" is not the appropriate word here. I would write "clear" instead.

**Reply:** Thank you for your suggestion. The term "tell-tale" will be replaced with "clear" in the revised manuscript.

Line 234: In line 230, the authors state that AOM activity is not associated with significant pore water $Fe^{2+}$ and $Mn^{2+}$ peaks in Zone-iii, but in line 234 they state that Fe-Mn AOM occurs in Zone-iii. These two statements are contradictory.

**Reply:** In line 230, we meant to convey that from the $CH_4$, DIC concentrations, and carbon isotope ratio of $CH_4$ and DIC, there is evidence of AOM taking place at those specific depths. AOM activity is identified by simultaneous depletion in $CH_4$ concentrations (1848.03 to 403.8 µM) and C-isotope ratios of $CH_4$ ($\delta^{13}C_{CH4}$: -106.39 to -73.63 ‰) and DIC ($\delta^{13}C_{DIC}$: -22.676 to -23.609 ‰) (Riedinger et al., 2014; Yoshinaga et al., 2014; Egger et al., 2016a,b; Luo et al., 2020; Aromokeye et al., 2020; Wegener et al., 2021). Since these AOM zones are also accompanied by porewater $Fe^{2+}$-$Mn^{2+}$ increase, we attribute AOM to Fe-Mn reduction (Luo et al., 2020). According to Fe-AOM stoichiometry, for every mole of DIC produced via Fe-AOM, 8 moles of $Fe^{2+}$ are produced. The DIC concentration perturbations (along the blue dashed lines) may be converted to equivalent µM of $Fe^{2+}$ produced via $Fe^{3+}$ reduction in the pore waters. The stoichiometrically calculated amount of $Fe^{2+}$ produced ranges from 11673 µM to 17163 µM in zone-i-ii and 6192 to 8259 µM in zone-iii. The measured porewater $Fe^{2+}$ concentrations are approximately 1.65-3% of the calculated $Fe^{2+}$ in zone-i-ii and 0.2-0.6 % in zone-iii. Thus, the measured $Fe^{2+}$ concentration is the residual $Fe^{2+}$ left after sulfidization

(FeS/FeS$_2$: Hensen et al., 2003; Treude et al., 2014; Peketi et al., 2015). Therefore, the lack of significant Fe$^{2+}$ and Mn$^{2+}$ enrichment in zone-iii may indicate complete consumption of Fe$^{2+}$ and Mn$^{2+}$ via sulfidization.

Line 278-279: CH$_4$ concentrations are generally low above the SMTZ (Otherwise there would be no SMTZ) and not only in the layers with Fe and Mn peaks.

**Reply:** We agree with the reviewer that above the SMTZ, the low CH$_4$ concentrations and an increase in DIC concentration are because of methanotrophy and the absence of methanogenesis. However, we have attributed the Fe-Mn enrichments in this zone to Fe-AOM mainly because of its covariation with DIC concentration and $\delta^{13}C_{DIC}$ values (Riedinger et al., 2014; Egger et al., 2016a,b; Li et al., 2019; Luo et al., 2020) similar to what observed below SMTZ.

Line 357-359: The last sentence is too general. Furthermore, SPOM is not the focus of this study and has only been mentioned once before. In the concluding sentence, the importance of the present study should be emphasized.

**Reply: We agree with the reviewer. We will be deleting the SPOM component from the text and only the main conclusion and significance of the study is retained.** The comprehensive geochemical investigation of the sediment core (5.25 m) retrieved from the seasonally hypoxic shelf sediment off the eastern Arabian Sea (west coast of India) has unveiled compelling evidence of focused Fe-Mn-AOM occurring at multiple depths beneath the seafloor. The accumulation of porewater Fe$^{2+}$ and Mn$^{2+}$, despite elevated concentrations of $\Sigma HS^-$, suggests the predominant production of Fe$^{2+}$ and Mn$^{2+}$ rather than consumption through metal sulfidization processes. This study demonstrates that Fe-Mn-AOM remains a significant biogeochemical phenomenon even within SO$_4^{2-}$ and $\Sigma HS^-$-rich sediments. Our findings are also relevant in assessing the consumption of CH$_4$, a greenhouse gas in organic-rich coastal sediments, thus having implications for climate change research. This study presents a novel perspective by documenting the biogeochemical heterogeneity in the occurrence of Fe-Mn-AOM, possibly attributed to deposition-controlled Fe-Mn reducing microbial population distribution in a highly dynamic coastal environment sensitive to climate change. Our results clearly call for the examination of the microbial communities in seasonally hypoxic sediment at high depth/temporal resolution, which may unravel the complex interactions between microbial life and the environment. The findings of the present study may have a far-reaching

influence on coupled $CH_4$-Mn-Fe-S cycling in expanding hypoxic coastal regions of the global ocean.

**Figures**

Figure 1b: What does the red line at the bottom right mean/indicate?

**Reply:** The red line in Fig. 1b in manuscript represents the Mandovi estuary

Figure 2: The figure is compressed. Labels a) to e) are not the same size and not at the same height.

**Reply:** Thank you for pointing that out. We uploaded a properly scaled file, however, the merged file became compressed. The revised figure will be included in the updated manuscript.

Figure 3: Can these two figures be placed side by side?

**Reply:** Thank you for the suggestion. The adjustment will be included in the revised manuscript.

Figure 4: The figure is also compressed. The TOC profile could also be added here.

**Reply:** Thank you for the valuable feedback. We will address the compression issue in Figure 4 and add the TOC profile to provide a more comprehensive view of the data. These changes will be incorporated into the revised version of the manuscript.

Figure 5a: The Fe monosulfide and pyrite content is also shown schematically in this figure. Is it assumed to be constant with depth (see also General Comment 4)?

**Reply:** The arrow does not indicate pyrite content, it only indicates the direction of pyrite burial.

Figure 5c: The $CH_4$ concentrations in the schematic representation reach near-zero values above the SMTZ (which is reasonable). How does Fe-Mn-AOM occur without any $CH_4$ above the SMTZ?

**Reply:** Above the SMTZ, the $CH_4$ concentrations range from 68.28 to 364.14 µM. Since it is significantly low compared to that below the SMTZ, we are unable to show that in the schematic diagram. We will be including the concentration ranges above the SMTZ in the schematic diagram in the revised manuscript to show that the values are non-zero. Moreover, the low $CH_4$ concentrations above the SMTZ are also associated with high DIC concentration

and low $\delta^{13}C_{DIC}$ values along with dissolved Fe-Mn enrichments which may be attributed to AOM (Riedinger et al., 2014; Egger et al., 2016a, b; Egger et al., 2017; Li et al., 2019; Luo et al., 2020).

**References**

Aromokeye, D. A., Kulkarni, A. C., Elvert, M., Wegener, G., Henkel, S., Coffinet, S., Eickhorst, T., Oni, O. E., Richter-Heitmann, T., and Schnakenberg, A.: Rates and microbial players of iron-driven anaerobic oxidation of methane in methanic marine sediments, Frontiers in Microbiology, 10, 3041, 2020

Antler, G., Mills, J. V., Hutchings, A. M., Redeker, K. R., and Turchyn, A. V.: The sedimentary carbon-sulfur-iron interplay–A lesson from East Anglian salt marsh sediments, Frontiers in Earth Science, 7, 140, 2019.

Berner, R. A.: Sulphate reduction, organic matter decomposition and pyrite formation, Philosophical Transactions of the Royal Society of London. Series A, Mathematical and Physical Sciences, 315, 25-38, 1985.

Bhattacharya, S., Mapder, T., Fernandes, S., Roy, C., Sarkar, J., Rameez, M. J., Mandal, S., Sar, A., Chakraborty, A. K., and Mondal, N.: Sedimentation rate and organic matter dynamics shape microbiomes across a continental margin, Biogeosciences, 18, 5203-5222, 2021.

Canfield, D. E., Raiswell, R., Westrich, J. T., Reaves, C. M., and Berner, R. A.: The use of chromium reduction in the analysis of reduced inorganic sulfur in sediments and shales, Chemical geology, 54, 149-155, 1986.

Canfield, D. E.: Reactive iron in marine sediments, Geochimica et Cosmochimica Acta, 53, 619-632, 1989.

Chambers, R. M., Hollibaugh, J., Snively, C. S., and Plant, J. N.: Iron, sulfur, and carbon diagenesis in sediments of Tomales Bay, California, Estuaries, 23, 1-9, 2000.

Egger, M., Lenstra, W., Jong, D., Meysman, F. J. R., Sapart, C. l. J., Van der Veen, C., Röckmann, T., Gonzalez, S., and Slomp, C. P.: Rapid sediment accumulation results in high methane effluxes from coastal sediments, PloS one, 11, e0161609, 2016a.

Egger, M., Kraal, P., Jilbert, T., Sulu-Gambari, F., Sapart, C. J., Röckmann, T., and Slomp, C. P.: Anaerobic oxidation of methane alters sediment records of sulfur, iron and phosphorus in the Black Sea, Biogeosciences, 13, 5333-5355, 2016b.

Egger, M., Hagens, M., Sapart, C. l. J., Dijkstra, N., van Helmond, N. A. G. M., Mogollón, J. M., Risgaard-Petersen, N., van der Veen, C., Kasten, S., and Riedinger, N.: Iron oxide reduction in methane-rich deep Baltic Sea sediments, Geochimica et Cosmochimica Acta, 207, 256-276, 2017.

Ferdelman, T. G.: The distribution of sulfur, iron, manganese, copper, and uranium in a salt marsh sediment core as determined by a sequential extraction method, University of Delaware, 1988.

Fernandes, S., Mazumdar, A., Bhattacharya, S., Peketi, A., Mapder, T., Roy, R., Carvalho, M. A., Roy, C., Mahalakshmi, P., and Da Silva, R.: Enhanced carbon-sulfur cycling in the sediments of Arabian Sea oxygen minimum zone center, Scientific reports, 8, 8665, 2018.

Fernandes, S., Mazumdar, A., Peketi, A., Anand, S. S., Rengarajan, R., Jose, A., Manaskanya, A., Carvalho, M. A., and Shetty, D.: Sulfidization processes in seasonally hypoxic shelf sediments: a study off the West coast of India, Marine and Petroleum Geology, 117, 104353, 2020.

Hensen, C., Zabel, M., Pfeifer, K., Schwenk, T., Kasten, S., Riedinger, N., Schulz, H., and Boetius, A.: Control of sulfate pore-water profiles by sedimentary events and the significance of anaerobic oxidation of methane for the burial of sulfur in marine sediments, Geochimica et Cosmochimica Acta, 67, 2631-2647, 2003.

Jørgensen, B. B.: Mineralization of organic matter in the sea bed—the role of sulphate reduction, Nature, 296, 643-645, 1982.

Jørgensen, B. B., Findlay, A. J., and Pellerin, A.: The biogeochemical sulfur cycle of marine sediments, Frontiers in Microbiology, 10, 849, 2019.

Li, J., Li, L., Bai, S., Ta, K., Xu, H., Chen, S., Pan, J., Li, M., Du, M., and Peng, X.: New insight into the biogeochemical cycling of methane, S and Fe above the Sulfate-Methane Transition Zone in methane hydrate-bearing sediments: A case study in the Dongsha area, South China Sea, Deep Sea Research Part I: Oceanographic Research Papers, 145, 97-108, 2019.

Lovley, D. R. and Phillips, E. J. P.: Organic matter mineralization with reduction of ferric iron in anaerobic sediments, Applied and environmental microbiology, 51, 683-689, 1986.

Luo, M., Torres, M. E., Hong, W.-L., Pape, T., Fronzek, J., Kutterolf, S., Mountjoy, J. J., Orpin, A., Henkel, S., and Huhn, K.: Impact of iron release by volcanic ash alteration on carbon cycling in sediments of the northern Hikurangi margin, Earth and Planetary Science Letters, 541, 116288, 2020.

Markovic, S., Paytan, A., and Wortmann, U. G.: Pleistocene sediment offloading and the global sulfur cycle, Biogeosciences, 12, 3043-3060, 2015.

Malinverno, A. and Pohlman, J. W.: Modeling sulfate reduction in methane hydrate-bearing continental margin sediments: Does a sulfate-methane transition require anaerobic oxidation of methane?, Geochemistry, Geophysics, Geosystems, 12, 2011.

Mazumdar, A., Peketi, A., Dewangan, P., Badesab, F., Ramprasad, T., Ramana, M. V., Patil, D. J., and Dayal, A.: Shallow gas charged sediments off the Indian west coast: Genesis and distribution, Marine Geology, 267, 71-85, 2009.

Mazumdar, A., Peketi, A., Joao, H., Dewangan, P., Borole, D. V., and Kocherla, M.: Sulfidization in a shallow coastal depositional setting: Diagenetic and palaeoclimatic implications, Chemical Geology, 322, 68-78, 2012.

Mehra, O. and Jackson, M.: Iron oxide removal from soils and clays by a dithionite-citrate system buffered with sodium bicarbonate, Clays and clay Minerals, 7, 317-327, 1960.

Naqvi, S., Jayakumar, D., Narvekar, P., Naik, H., Sarma, V., D'Souza, W., Joseph, S., and George, M.: Increased marine production of N2O due to intensifying anoxia on the Indian continental shelf, Nature, 408, 346-349, 2000.

Naqvi, S. W. A., Naik, H., Jayakumar, D., Shailaja, M., and Narvekar, P.: Seasonal oxygen deficiency over the western continental shelf of India, Past and present water column anoxia, 195-224, 2006.

Peketi, A., Mazumdar, A., Joao, H. M., Patil, D. J., Usapkar, A., and Dewangan, P.: Coupled C-S-Fe geochemistry in a rapidly accumulating marine sedimentary system: diagenetic and depositional implications, Geochemistry, Geophysics, Geosystems, 16, 2865-2883, 2015.

Raiswell, R. and Berner, R. A.: Pyrite formation in euxinic and semi-euxinic sediments, American Journal of Science, 285, 710-724, 1985.

Raiswell, R., Vu, H. P., Brinza, L., and Benning, L. G.: The determination of labile Fe in ferrihydrite by ascorbic acid extraction: methodology, dissolution kinetics and loss of solubility with age and de-watering, Chemical Geology, 278, 70-79,

Raven, M. R., Sessions, A. L., Adkins, J. F., and Thunell, R. C.: Rapid organic matter sulfurization in sinking particles from the Cariaco Basin water column, Geochimica et Cosmochimica Acta, 190, 175-190, 2016.

Rickard, D. and Luther, G. W.: Chemistry of iron sulfides, Chemical reviews, 107, 514-562, 2007.

Riedinger, N., Formolo, M. J., Lyons, T. W., Henkel, S., Beck, A., and Kasten, S.: An inorganic geochemical argument for coupled anaerobic oxidation of methane and iron reduction in marine sediments, Geobiology, 2014.

Roden, E. E. and Wetzel, R. G.: Competition between Fe (III)-reducing and methanogenic bacteria for acetate in iron-rich freshwater sediments, 2003.

Schott, F. A. and McCreary Jr, J. P.: The monsoon circulation of the Indian Ocean, Progress in Oceanography, 51, 1-123, 2001.

Sebastian, T., Nath, B. N., Naik, S., Borole, D. V., Pierre, S., and Yazing, A. K.: Offshore sediments record the history of onshore iron ore mining in Goa State, India, Marine pollution bulletin, 114, 805-815, 2017.

Treude, T., Krause, S., Maltby, J., Dale, A. W., Coffin, R., and Hamdan, L. J.: Sulfate reduction and methane oxidation activity below the sulfate-methane transition zone in Alaskan Beaufort Sea continental margin sediments: Implications for deep sulfur cycling, Geochimica et Cosmochimica Acta, 144, 217-237, 2014.

Wilkin, R. T. and Barnes, H. L.: Formation processes of framboidal pyrite, Geochimica et Cosmochimica Acta, 61, 323-339, 1997.

Werne, J. P., Lyons, T. W., Hollander, D. J., Formolo, M. J., and Damsté, J. S. S.: Reduced sulfur in euxinic sediments of the Cariaco Basin: sulfur isotope constraints on organic sulfur formation, Chemical geology, 195, 159-179, 2003.

Wegener, G., Gropp, J., Taubner, H., Halevy, I., and Elvert, M.: Sulfate-dependent reversibility of intracellular reactions explains the opposing isotope effects in the anaerobic oxidation of methane, Science Advances, 7, eabe4939, 2021.

Whiticar, M. J.: Carbon and hydrogen isotope systematics of bacterial formation and oxidation of methane, Chemical Geology, 161, 291-314, 1999.

Yoshinaga, M. Y., Holler, T., Goldhammer, T., Wegener, G., Pohlman, J. W., Brunner, B., Kuypers, M. M. M., Hinrichs, K.-U., and Elvert, M.: Carbon isotope equilibration during sulphate-limited anaerobic oxidation of methane, Nature Geoscience, 7, 190-194, 2014.

Yücel, M., Konovalov, S. K., Moore, T. S., Janzen, C. P., & Luther III, G. W.: Sulfur speciation in the upper Black Sea sediments, Chemical geology, 269, 364-375, 2010.

Zhu, M. X., Chen, K. K., Yang, G. P., Fan, D. J., and Li, T.: Sulfur and iron diagenesis in temperate unsteady sediments of the East China Sea inner shelf and a comparison with tropical mobile mud belts (MMBs), Journal of Geophysical Research: Biogeosciences, 121, 2811-2828, 2016

---

## Author Comment (AC2)

**Response to comments of Reviewer 2**

**Reviewer 2**

This manuscript presents geochemical data for sediments from a long core from the Eastern Arabian Sea. The authors combine porewater profiles of methane, sulfide, $Fe^{2+}$ $Mn^{2+}$, and DIC with isotopic data for methane and DIC and results of sediment Fe speciation (2 steps). The results are used to argue that the anaerobic oxidation of methane is coupled to Fe and Mn oxide reduction in a series of discrete layers in the sediment. While the topic is important and of interest to the BG readership, I have many concerns about the data and interpretation, as detailed below.

Methods:

Comment 1: Iron and manganese in porewaters and sediments are highly sensitive to oxidation artifacts. The depth trends of both dissolved Fe and Mn show a range of spikes that are typical for profiles with such artefacts. The procedure for porewater collection and processing described in the methods refers to "a stream of argon gas" used to avoid oxidation of dissolved sulfide. The measures taken to avoid oxidation of dissolved Fe and Mn are not described. Notably, such a stream of gas does not generally prevent $Fe^{2+}$ from oxidation. This is why sediment and porewater samples used for $Fe^{2+}$ and $Mn^{2+}$ analyses should be shielded fully from the atmosphere (e.g. in a glove box filled with argon or nitrogen) until the porewater samples are placed into in the vial in which the acid will be added to keep them in solution. Furthermore, no measures to avoid oxidation of the solid phase samples are mentioned in the text. FeS can easily oxidize to Fe oxides upon brief exposure to oxygen. The authors should clarify what they did and address the implications for their results.

**Reply:** Thanks for your comments. We acknowledge the importance of mitigating such artifacts in our analyses. To address these concerns, we ensured that proper sampling protocol was followed during porewater extraction and processing.

(A) The sediment core collection

The reviewer has expressed doubt regarding the oxidation of the porewater by atmospheric $O_2$ during sampling. The reviewer has raised certain issues regarding the validity of Fe

concentration measurements in the porewater and the importance of nanoparticles, especially Fe-Mn sulfides.

It has been mentioned in our methodology that solid phase sampling and preparation of various aliquots of porewater were all carried out under an argon gas head. The typical experimental setup which we have been using for several years (Mazumdar et al., 2012; Fernandes et al., 2018; Fernandes et al., 2020) for onboard porewater work is represented schematically (Figure 1). In this experimental protocol, argon gas is allowed to fall on the centrifuge tube where an argon atmosphere is created above and around the tube where no ambient oxygen is allowed into the system. This process does not allow diffusion of oxygen into the porewater. In addition, this entire set up is hosted inside a sampling box with transparent clear PVC curtain strips. Our previous experience with this setup (references cited above) successfully shows an oxidation-free sampling protocol. Since we are handling a 6 m long core, working within a glove bag is not feasible. Reported $H_2S$ concentrations in this core range up to 10 mM. Oxidation of $H_2S$ may influence the $SO_4^{2-}$ concentration and $\delta^{34}S_{SO42-}$ values (Figure 2C) which is not apparent in the porewater profiles. In Fernandes et al. (2020), two cores from the shelf region are studied using the same protocol, however, the $SO_4^{2-}$ concentration and isotope profile do not show any evidence of sulfide oxidation. Mazumdar et al. (2012) (Chemical Geology_supplementary text) carried out porewater $\delta^{18}O_{SO42-}$ measurements to investigate possible contamination of porewater sulfate by sulfide oxidation with atmospheric oxygen. However, the results showed no evidence of sulfide oxidation. Hopefully, the reviewer will accept our observations. We will include these sampling details in the revised text and supplementary material.

[Figure]

*Figure 1: Schematic diagram showing the sampling protocol followed for porewater extraction followed in the present study*

[Figure]

*Figure 2: (A) Porewater $SO_4^{2-}$ (mM) and $\Sigma HS^-$ (μM) concentrations, (B) Dissolved Fe and Mn (μM) concentrations, and (C) sulfur isotope ratio of $SO_4^{2-}$ ($\delta^{34}S_{SO42-}$) and $HS^-$ ($\delta^{34}S_{HS^-}$).*

B) Nanoparticle formation and influence on iron concentration

We deeply appreciate the reviewer for raising this very pertinent issue regarding iron sulfide nanoparticles in the porewater. We agree that we should have discussed this in the manuscript. At high hydrogen sulfide concentrations, $Fe^{2+}$ is likely to be present in the porewater as $FeS_{nano}$

(Matamoros-Veloza et al., 2018), mackinawite or any other stable Fe-S nanoparticle form (Rickard and Morse, 2005; Rickard and Luther, 2007). We believe that $Fe^{2+}$ is generated by $Fe^{3+}$ reduction via the AOM process. The $Fe^{2+}$ required for the formation of the FeS nanoparticles may be produced via the Fe-AOM pathway. The nanoparticles will pass through 0.22-micron syringe filters and eventually be in the aliquots for metal concentration measurement. The FeS/MnS nanoparticles will dissolve in supra pure nitric acid and will be measured as the total Fe concentration in the pore water. It is worth noting that except for the Fe concentration spikes, the background Fe concentrations range from 9.05 to 43.89 μM compared to the spike values from 164.94 to 387.54 μM (Figure 3D). It is also apparent that the Fe spikes are strictly associated with the AOM zones identified by methane concentrations, methane carbon isotope ratios, DIC concentrations, and DIC carbon isotope ratios (Figure 3A-E). It may be noted that the $H_2S$ is high throughout the core below 63 cmbsf. Had it been a case of artifact generation, $Fe^{2+}$ spikes wouldn't have been restricted to the AOM zones.

[Figure]

**Figure 3**: *Figure showing Fe-Mn-AOM and $SO_4^{2-}$-AOM activities by means of pore fluid and solid phase chemistry in seasonally hypoxic coastal sediment of the Eastern Arabian Sea. (A) Porewater $SO_4^{2-}$ (mM) and $\Sigma HS^-$ (μM) concentrations. (B) Dissolved $CH_4$ concentrations (μM) and $\delta^{13}C_{CH4}$ values. (C) Dissolved DIC concentrations (mM) and*

$\delta^{13}C_{DIC}$ *values (D, E, F) Dissolved Fe and Mn (mM) concentrations. The blue dashed lines represent the depth layers with AOM activities. The red curly bracket depicts SMTZ. The different color shades represent zone-i, zone-ii, and zone-iii.*

C)     Recent publication on Fe-Mn-AOM (Li et al., 2019; Deep-Sea Research, Part-I) have reported H₂S concentrations of up to 10 mM in a 3.3 m long core from the Dongsha area. Within the Fe-AOM zone, Li et al. (2019) have reported $Fe^{2+}$ concentrations up to 50 µM and interpreted it as a product of Fe-AOM. They have also reported depletion of DIC isotope ratio up to -30 ‰ and attributed it to the AOM process. It is important to note that Li et al. (2019) have used the Rhizon sampler to avoid atmospheric exposure. The coexistence of $Fe^{2+}$ with concentration varying from 25 to 50 µM along with H₂S concentration of 1 to 3 mM is also reported in Knab et al. (2009) in Black Sea sediments.

D)  Solid phase sampling

The sediment coring was carried out using a predrilled PVC core liner. The center-to-center distance of the drill holes was maintained at 10 cm to avoid any kind of interference during sampling. Sampling was carried out using gas-tight 50 ml cut syringes. The sediments were extracted under multiple argon flows and stored in 50 ml pp argon flushed centrifuge tubes to the brink to avoid any kind of headspace. The centrifuge tubes were immediately sealed with teflon tapes and centrifuged for porewater extraction. Please note that we are handling 6 m long sediment cores. So, it's very difficult to work in a glove bag.

Comment 2: Methane is known to degas from sediments upon sample retrieval (see, for example: Jorgensen, 2021. Geochemical Perspectives 10 (2)) The potential impact of this process on the methane profile and other results should be discussed.

**Reply:** Thank you for pointing out this important aspect. We acknowledge that methane degassing can occur during sediment retrieval. However, a comparative study on methane degassing during core retrieval using a pressure core sampler and conventional coring procedures (standard advanced piston (APC) and extended core barrel (XCB)) by Wallace et al. (2000) has shown that insignificant isotopic fractionation happens if there is a methane loss due to methane degassing while coring.

The methane loss during core recovery depends on pressure drop relative to the sea bed and the time gap between coring and sampling. In our case, the pressure drop is up to 3 atm at 30

m water depth and the temperature is 25° C, at this condition, the gas loss is minimal (Sivan et al., 2011). Immediately after core retrieval, for headspace methane analysis, the sediment was extracted using 50 ml cut syringes at an interval of 10 cm and transferred into 20 ml helium flushed headspace vials filled with 3 ml of KOH and 3 ml of $NaN_3$ to trap $CO_2$ and stop the microbial activity. The vials were homogenized (vigorously shaken), inverted, and stored at 4°C after sealing with butyl rubber septa.

**Presentation and interpretation**

Comment 3: The presentation and discussion of the results could be done in a much more structured and balanced way and many of the conclusions are speculative. The combination of the results and discussion makes it hard to obtain an overview of the data. Many interpretations in the text do not appear to be fully supported by the data. For example, the separation into "zones" with different "diagenetic regimes" based on porewater data alone is rather arbitrary. A firm case for Fe- and Mn-AOM in a series of specific zones (which is highly unusual!) requires strong support from solid phase and microbial data for the same sediment intervals and appropriate stoichiometric calculations and a scenario with a timeline of deposition and diagenesis that can also explain the generation of the spikes in Fe and Mn. Such data and such a scenario are not provided in the manuscript. In fact, based on the data presented, I don't see a clear case for Fe-AOM or Mn-AOM.

**Reply:** Thank you for your suggestion. We will carefully consider this revision and reorganize the manuscript to improve the overall understanding.

A) Zonation in the sediment

We have carried out complete grain size, porosity, clay mineralogy, organic carbon content, and $\delta^{13}C_{TOC}$ analysis (all presented here and will be given in the revised manuscript). The sediments of this region are clayey silt with median grain size (D50) showing a small variability 6.66 to 17.43 um (Figure 4D). The clay mineralogical studies show a uniform mineralogy composed of Kaolinite and montmorillonite with low levels of illite. The zones are homogenous in terms of porosity (Figure 4A) except for some spikes in between. The $Fe^{2+}/Mn^{2+}$ enrichment zones do not show any correlation with TOC content, TOC/TN$_{(molar)}$, and $\delta^{13}C_{TOC}$ values (Figure 4). None of the vertical profiles help in zonation of the sediment which can correlate with the observed porewater profiles. The sedimentological characteristics in the core don't show any obvious correlation with the biogeochemical observations. Therefore, the

physical parameters are not responsible for driving the Fe-AOM in our study area. In our view, the zonation is only possible with the porewater profile.

[Figure]

*Figure 4: Depth profiles of (A) porosity, porewater (B) Fe²⁺ (C) Mn²⁺ (D) median grain size (D50) (E) TOC (mg/g) (F) TOC/TN(molar) and δ¹³C_TOC (‰ VPDB) in the present study (SSD070/7/GC6).*

B) Microbiology data

The generation of microbiology data is beyond the scope of the present manuscript. However, we shall be correlating with microbiologists in our future expeditions in this region.

C) Regarding speculative statements

Regarding microbiology, we haven't made any speculative statements. We have only suggested that a microbiological study has to be carried out in this region and the statements included are

"This study presents a new perspective by documenting the biogeochemical heterogeneity in the occurrence of Fe-Mn-AOM, possibly attributed to deposition-controlled Fe-Mn reducing microbial population distribution in a highly dynamic coastal environment sensitive to climate change. The findings of the present study may have a far-reaching influence on coupled CH₄-Mn-Fe-S cycling in expanding hypoxic coastal regions of the global ocean. Our findings lead to a hypothesis that necessitates future examination of microbial communities in seasonally hypoxic sediments and SPOM at high depth and temporal resolution which may uncover the complex interactions between microbial life and their environment."

We have done stoichiometric calculations with DIC perturbations and is also written in the text.

Comment 4: Various terms are used that are not well-defined, such as "vital", "tell-tale", "biogeochemical phenomenon" etc

**Reply:** Thank you for pointing out the need for clearer definitions of the terms used in our manuscript. To address this, we will revise the manuscript to either define or replace these terms with more specific language.

Detailed comments:

Comment 5: Line 13: the authors write that metal-driven AOM is a "globally important biogeochemical process….". To my knowledge, such a global role has so far not been shown.

**Reply:** We appreciate the reviewer's comment on the global significance of metal-driven anaerobic oxidation of methane (AOM). Our statement aimed to highlight the potential impact of this process in various marine environments where metal oxides, particularly iron (Fe) and manganese (Mn), play a crucial role in AOM. While it is true that the global significance of metal-driven AOM is still underexplored, recent studies suggest that this process may contribute significantly to methane cycling in marine sediments (Riedinger et al., 2014; Egger et al., 2015, 2016a, b, 2017; Aromokeye et al., 2020; Xiao et al., 2023). Iron-driven AOM (Fe-AOM) rates have indeed been reported to be generally low, which may be due to the challenges microbes face in accessing solid iron oxides as electron acceptors (Lalonde et al., 2012). For example, radiotracer experiments at the Helgoland Mud Area have demonstrated Fe-AOM rates of $0.095 \pm 0.03$ nmol cm$^{-3}$ d$^{-1}$, which is approximately 2% of the rate observed for $SO_4^{2-}$-AOM in the sulfate-methane transition (SMT) zone (Aromokeye et al., 2020). Similarly, modeled estimates from the coastal sediments of the Bothnian Sea suggest that Fe-AOM contributes about 3% compared to the 97% contribution of $SO_4^{2-}$-AOM to overall methane consumption (Egger et al., 2015). These findings indicate that while Fe-AOM rates are lower than those of $SO_4^{2-}$-AOM, they still represent a significant methane sink in Fe-rich environments. **Given these considerations, we will revise our statement to more accurately reflect the current understanding of Fe-AOM's role.**

Comment 6: Line 18: from the data presented, the conclusion that there is Fe-Mn-AOM is speculative. Further evidence should be provided, including additional solid phase data and

porewater profiles that allow a case to be made that the spikes are real and not oxidation artifacts.

**Reply:** In response to review question no.1, we have elaborately shown that our porewater sampling method has no scope for oxidation. The observed exact correspondence of $Fe^{2+}/Mn^{2+}$ concentration with AOM zones (increase in DIC concentration, depleted $\delta^{13}C_{DIC}$ and $\delta^{13}C_{CH4}$ values) suggests that AOM (Figure 3) is primarily controlled by Fe-Mn reduction (Luo et al., 2020). We have not speculated anything in the conclusion. We have only suggested the possibility of microbial role in driving focussed Fe-Mn-AOM in the present study and may be investigated in future studies, which we will be carrying out in future expeditions. To gather additional porewater data, we have to carry out another research cruise which is not possible right now. We will be carrying out higher-resolution fluid chemistry, solid phase chemistry, and microbiological studies along with the microbiology group in future expeditions. Additional solid phase data (sedimentological characteristics) have been provided in Figure 4 and discussed in response to review comment 3(A) which also do not show any obvious correlation with the observed porewater profiles. **However, we will include this aspect suggested by the reviewer in the revised text.**

Comment 7: Line 19. Change to in the sediment, not in the core

**Reply:** Thank you so much for the comment. The term "core" will be changed to "sediment" in the revised manuscript.

Comment 8: Line 21, 22. Here, an increase in delta $^{13}C_{CH4}$ would be expected.

**Reply:** Thank you for your observation regarding the expected increase in $\delta^{13}C$-$CH_4$ during anaerobic oxidation of methane (AOM). Many studies have reported minima in $\delta^{13}C$ values of both DIC and $CH_4$ during AOM (e.g., Borowski et al., 2001; Knab et al., 2009; Sauer et al., 2015; Geprägs et al., 2016; Luo et al., 2020). Anaerobic oxidation of methane associated with $SO_4^{2-}$-AOM or Fe-AOM results in depleted $\delta^{13}C_{CH4}$ and $\delta^{13}C_{DIC}$ due to the following factors (i) the recycling of $^{13}C$-depleted DIC produced by AOM back to $CH_4$ through $CO_2$ reduction (Borowski et al., 1997), (ii) microbe-mediated carbon isotope fractionations between forward and backward AOM reactions at low sulfate concentrations (Holler et al., 2011; Yoshinaga et al., 2014) and (iii) intracellular reaction reversibilities along enzymatic AOM pathway (Wegener et al., 2021). Egger et al. (2017) and Luo et al. (2020) have shown a decrease in

$\delta^{13}C_{CH4}$ and $\delta^{13}C_{DIC}$ values during AOM. Therefore, negative excursions of $\delta^{13}C$ values of both DIC and $CH_4$ at Fe-Mn-AOM depths are important evidence for AOM in the present study.

**Comment 9:** Line 25: Microbes follow substrates, so this is rather speculative without data to support this.

**Reply:** Unfortunately, we do not have microbiological data available to support this specific observation. However, the geochemical data strongly confirms the occurrence of Fe-Mn-AOM at multiple depths below the seafloor. Here the co-variations in the concentrations and carbon isotopic values of $CH_4$ and DIC occur at multiple depths concomitant with an increase in $Fe^{2+}$ and $Mn^{2+}$ concentrations (Figure 3). Given that the necessary substrates for Fe-Mn-AOM-such as $CH_4$ and reactive Fe-oxy(hydr)oxides-are present, we have only suggested that microbiological studies may be carried out in future expeditions.

**Comment 10:** Line 29: If there are far-reaching implications, please specify them.

**Reply:** Thank you for the comment. We will remove the term in the revised manuscript.

**Comment 11:** Line 32: "a vital process" => what does "vital" refer to?

**Reply:** The vital word will be replaced by the word "important" in the revised manuscript.

**Comment 12:** Line 33-35: Please be more specific about the impact on global cycles or remove it.

**Reply:** We will remove the term in the revised manuscript.

**Comment 13:** Line 38: Better to remove "syntrophic" since ANMEs may also do this alone as mentioned later in the text.

**Reply:** Thank you for your suggestion. We agree that the term "syntrophic" may not be necessary in this context, as ANMEs (anaerobic methane-oxidizing archaea) are capable of performing these processes independently, as discussed later in the text. We will remove "syntrophic" from Line 38.

**Comment 14:** Line 54: Please be more specific about the type of kinetics.

**Reply:** Although Fe-Mn-AOM is thermodynamically more favorable than $SO_4^{2-}$-AOM, $SO_4^{2-}$-AOM is responsible for > 90% of global methane consumption because $SO_4^{2-}$ is present in its ionic form (Beal et al., 2009; Knittel and Boetius, 2009). The crystallinity and conductivity of iron oxides significantly influence Fe-AOM in iron oxide-rich sediments. The limited

availability of Fe(III) oxides for reduction is attributed to the solid Fe(III) form and the slow electron transfer between the solid phase and the organic acceptor (Lalonde et al., 2012). Lovley and Phillips (1986) demonstrated that while amorphous Fe(III) oxyhydroxides are easily reduced by microbes, magnetite, mixed Fe(III)–Fe(II) compounds, and most oxalate-extractable Fe(III) minerals are not readily available for microbial reduction. Experimental results from Norði et al. (2013) indicated that iron-dependent AOM is more energetically favorable with amorphous ferric oxyhydroxide than with crystalline goethite as the electron acceptor. It is suggested that the kinetic limitations, crystalline structure, or surface charge alterations due to ion adsorption make it more challenging for crystalline Fe-oxides to undergo biological reduction by methane (Lovley and Phillips, 1988; Hori et al., 2015).

Comment 15: Line 62: This is more complex than described: sulfate reduction coupled to organic matter degradation can generate sulfide that can react with the Fe and Mn oxides

**Reply:** Thank you for this observation. We agree that sulfate reduction coupled with organic matter degradation can produce sulfide, which may react with Fe and Mn oxides. However, as demonstrated by Li et al. (2019), Fe- AOM can still occur within the sulfate reduction zone, even in the presence of significant $H_2S$ concentrations (3.69–12.71 mmol/L). The study observed elevated DIC concentrations and more negative $\delta^{13}C_{DIC}$ values, along with decreasing sulfate and Fe(III) levels that point to the occurrence of AOM above the sulfate-methane transition zone (SMTZ). Furthermore, high-throughput sequencing study also detected the presence of ANME-1 archaea and bacterial species such as *Sulfurovum* and *Shewanella*, which are involved in methane, sulfur, and iron cycling. Metagenomic analyses also revealed functional genes related to sulfate reduction, sulfur oxidation, and iron uptake, further supporting the interplay between these processes in the sediments. These observations are also supported by an incubation study (Segarra et al., 2013) where notable rates of Fe-Mn-AOM coincide with comparatively elevated levels of $SO_4^{2-}$-AOM.

Comment 16: Line 70: Sources of the data used for Figure 1B should be given

**Reply:** We will add the data for Figure 1B in the revised manuscript and supplementary file.

Comment 17: Line 71. Remove tell-tale

**Reply:** Thank you for your observation. We will remove the term "tell-tale" from Line 71 in the revised manuscript.

Comment 18: Line 73. Not clear what the number 1,80,000 refers to.

**Reply:** The number 180,000 km$^2$ refers to the area covered by severely hypoxic waters along the entire shelf of the west coast of India during southwest monsoon (september-october) (Figure 5) (Naqvi et al., 2000).

[Figure]

*Figure 5: Zone of severe hypoxia on the Western continental shelf of India covering an area of 1,80,000 km$^2$ (marked by shaded region) (Naqvi et al., 2000)*

Comment 19: Line 78. Add a comma before "respectively"

**Reply:** Thank you for pointing out that. The text will be revised accordingly.

Comment 20: Line 88. I would suggest to use the same marker for the study site in both figures

**Reply:** Thank you for the suggestion. We will revise the figure accordingly.

Comment 21: Line 93-94: I would propose to only add sites that are relevant to this study

**Reply:** Thank you for the suggestion. We will revise the figures accordingly to include only the sites relevant to the present study.

Comment 22: Line 109. Were the vials stored upside down?

**Reply:** For headspace methane analysis, the sediment was extracted using 50 ml cut syringes at an interval of 10 cm and transferred into 20 ml helium-flushed headspace vials filled with 3 ml of KOH and 3 ml of NaN$_3$ to trap CO$_2$ and stop microbial activities respectively. The vials

were homogenized (vigorously shaken), inverted, and stored at 4°C after sealing with butyl rubber septa.

Comment 23: Line 111-112 using a stream of gas is not sufficient to avoid oxidation artifacts

**Reply:** Please see the response to review question no.1 in which we have elaborately shown that our porewater sampling method has no scope for oxidation.

Comment 24: Line 113: add the material of the filter

**Reply:** Thank you for pointing that out. We used 0.22 um PTFE membrane syringe filters for the filtration process. We will add this detail to Line 113 in the revised manuscript.

Comment 25: Line 114 helium headspace

**Reply:** Thank you for pointing out that. The filtered porewater was crimp-sealed after flushing with argon and stored at 4°C until shore-based analysis. The text will be revised accordingly.

Comment 26: Line 116. Please add a reference for the method for sulfide trapping with Cd

**Reply:** Thank you for the suggestion. We will add the reference for the method of sulfide trapping with Cd as described by Fernandes et al. (2020) to Line 116.

Comment 27: Line 117: Please provide the amount of acid per volume added

**Reply:** Subsamples for total dissolved Fe and Mn were acidified with 100 μL of 35% Suprapure $HNO_3$.

Comment 28: Line 142: Please list the metals that are presented in the manuscript

**Reply:** Thank you for your comment. We will list the metals presented in the manuscript as dissolved Fe and Mn on Line 142.

Comment 29: This is sulfate and not the sum of $HS^-$

**Reply:** The total $HS^-$ was removed by adding $CdNO_3$ in order to avoid oxidation of sulfide which can change $SO_4^{2-}$ concentrations.

Comment 30: Line 152. The detection limit is relevant here and should be given.

**Reply:** Thank you for the suggestion. We will include the detection limit for $SO_4^{2-}$ analysis, which is 0.00005 mM, on Line 152.

Comment 31: Line 161. Explain what Fe minerals the asc and dithionite fractions are a measure of

**Reply:** The $Fe_{Asc}$ represents highly reactive/bioavailable ferric iron ($Fe_{Asc}$) particularly ferrihydrite (Hyacinthe et al., 2006; Raiswell et al., 2010; Riedinger et al., 2014) and dithionite fractions represent lepidocrocite, goethite, and hematite (Mehra and Jackson, 1960; Canfield et al., 1989).

Comment 32: Line 176. Please add how the sample was desalinated.

**Reply:** Thank you for your observation. We will add the following details on Line 176: The freeze-dried samples were washed with ultra-pure water (18 MΩ) several times, followed by centrifugation. The supernatant liquid was checked for salinity using $AgNO_3$ and washed with milliq several times until no milky precipitate of AgCl was formed. After desalination, the samples were dried, homogenized, and stored for total carbon (TC) and total inorganic carbon (TIC) measurements.

Comment 33: Line 188. Please focus on the sediment, not the core

**Reply:** Thank you for the suggestion. The suggested change will be incorporated in the revised manuscript.

Comment 34: Figure 2. It is unusual that $H_2S$ and $Fe^{2+}$ co-occur in porewaters since they would be expected to co-precipitate as FeS. The spikes in Fe and Mn are typical for oxidation artifacts. This warrants further research (e.g. repeated sampling at the same site while taking measures to fully exclude oxidation during sampling).

**Reply:** We have explained in review no: 1 that our sampling protocol is oxidation-free. Please see the schematic diagram of our experimental protocol (Figure 1) which we have been following many years in our previous publications. We have stated earlier that FeS nanoparticle formation is a possible pathway of stabilization of $Fe^{2+}$ in the porewater (Rickard and Luther, 2007; Olson et al., 2017). In the case of oxidation of FeS nanoparticles, they are likely to be converted to some ferric hydroxide forms. The partial and complete oxidation of FeS can be shown by the following equations (eq.1, 2, 3).

$$2FeS + O_2 \xrightarrow{2H_2O} 2Fe^{2+}(OH^-)_2 + 2S^0 \qquad \text{(eq.1) (Baikova et al. (2009)}$$

$$FeS + 1/2O_{2(aq)} + 2H^+ = S_{(s)} + Fe^{2+}_{(aq)} + H_2O \qquad \text{(eq.2) Chiriţă et al. (2008)}$$

$$FeS + 3/4O_{2(aq)} + 1/2H_2O = S_{(s)} + FeOOH_{(s)} \qquad \text{(eq.3) Chiriţă et al. (2008)}$$

Acid treatment of the oxidized porewater aliquots is unlikely to change the concentration of total Fe in porewater because oxidation of FeS will only change the form but the concentration of Fe remains the same. We have also stated that our sulfate concentration and $\delta^{34}S_{SO4^{2-}}$ profile (Figure 2C) do not show any evidence of additional sulfate formation from sulfide oxidation. An artifact generation is expected to be erratic irrespective of other geochemical parameters. However, in the present study, we have repeatedly emphasized that Fe-Mn spikes are only restricted to Fe-Mn-AOM zones and not erratically distributed all over the cores. We believe that this is convincing evidence against any oxidation artifact.

Comment 35: Line 209: It is critical to be able to exclude methane-degassing artifacts. Such spikes in methane are unusual and require large variations in methane production and removal over short depth intervals if the interpretation here is correct. Some quantification of the corresponding processes is needed to support the interpretation. The stoichiometry of Fe-AOM is such that you need quite some more change in dissolved $Fe^{2+}$ than is seen here.

**Reply:** Thank you for pointing out this important aspect. We acknowledge that methane degassing can occur during sediment retrieval.

A) Methane degassing

The methane loss during core recovery depends on pressure drop relative to the sea bed, permeability of sediment, and the time gap between coring and sampling. Methane loss in porewater during core recovery is primarily due to a decrease in solubility because of pressure drop. In our case, the core recovery is from 30 m water depth which leads to a pressure drop of 3 atm. Using the methane solubility graph (Behrouz and Aghajani, 2015), it may be noted that the drop in mole % is 0.00006 which is quite low and may not lead to significant gas loss. As mentioned in response to review comment no:3, the sediments of the present study are fine clay-silt rich with median grain size (D50) varying from 6.66 to 17.43 um.  Such fine-grain sediments have poor permeability which results in low methane loss. So, a 3°C rise in temperature relative to seabed, a low-pressure drop of 3atm, and low permeability will lead to minimum methane loss. Knab et al (2009) reported insignificant methane loss in their sediment core from the Black Sea. However, they also observed a 500 to 800 μM change in methane concentration below the SMTZ. Significant variation in methane concentration below the SMTZ has also been observed by several other studies such as Emeis et al. (2004), Li et al. (2012), Cho et al. (2015), and Yang et al. (2023).  Moreover, during Fe-AOM, Egger et al. (2017) and Luo et al. (2020) have shown a significant drop in $CH_4$ concentration and $\delta^{13}C_{CH_4}$

values. The drop in methane concentration and isotope ratio depends on the substrate and microbial diversity and activity.

B) Regarding low porewater $Fe^{2+}$ concentrations

Regarding the reviewer's comment on low $Fe^{2+}$ concentration, at high dissolved sulfide concentrations observed in the present study, a significant amount of $Fe^{2+}$ might have reacted with $H_2S$ to form iron monosulfide and pyrite. So, the measured $Fe^{2+}$ is the residual $Fe^{2+}$ left after sulfidization (FeS/$FeS_2$: Hensen et al., 2003; Treude et al., 2014; Peketi et al., 2015). Moreover, at low/negligible sulfide concentrations, previous studies (Vigderovich et al., 2019; Aromokeye et al., 2020; Luo et al., 2020) have reported Fe-AOM at dissolved iron concentrations (Figures 6, 7, 8) similar to that reported here.

[Figure]

*Figure 6: Porewater profiles depicting the existence of the Fe-AOM in the methanic sediments of Helgoland Mud Area (Aromokeye et al., 2020). Pore-water profiles of sulfate, sulfide, methane, dissolved iron, and dissolved manganese in the sediments.*

[Figure]

*Figure 7: Pore-water profiles of methane, $\delta^{13}C_{DIC}$ and dissolved iron depicting the existence of the Fe-AOM in the methanic sediments (Vigderovich et al., 2019).*

[Figure]

*Figure 8: Pore-water profiles of (A) $SO_4^{2-}$ and $H_2S$, (B) $CH_4$ and $\delta^{13}C_{CH4}$, (C) DIC and $\delta^{13}C_{DIC}$, (F) $Fe^{2+}$ in sediments of the northern Hikurangi margin (Luo et al., 2020).*

Comment 36: Line 212: Please clarify what is meant by "Fe-Mn-AOM specific points"

**Reply:** The Fe-Mn-AOM-specific points were written to represent the depths where Fe-Mn-AOM is observed.

Comment 37: Line 229 Please clarify what the basis is for the conclusion that there is AOM in these layers.

**Reply:** The Fe-Mn-AOM activity is identified by the simultaneous depletion in $CH_4$ concentrations, $\delta^{13}C_{CH4}$, and $\delta^{13}C_{DIC}$ values coupled with an increase in $Fe^{2+}$ and $Mn^{2+}$ concentrations (Riedinger et al., 2014; Egger et al., 2016a, b, 2017; Vigderovich et al., 2019; Luo et al., 2020; Aromokeye et al., 2020). The punctuated increase in $Fe^{2+}/Mn^{2+}$ concentrations observed below the seabed in the present study corresponds to an increase in DIC concentration and depletion in $CH_4$ concentrations, $\delta^{13}C_{CH4}$, and $\delta^{13}C_{DIC}$ values which in turn confirms the occurrence of Fe-Mn-AOM in those depth zones.

Comment 38: Line 232 See above. To the reader, it is not clear that there is evidence for Fe-Mn-AOM.

**Reply:** Following are the evidence for Fe-Mn-AOM

Metal-driven AOM are typically characterized by the following biogeochemical signatures (Riedinger et al., 2014; Egger et al., 2016a, b, 2017; Vigderovich et al., 2019; Li et al., 2019; Luo et al., 2020; Aromokeye et al., 2020; Xiao et al., 2023).

i) Significant depletion in methane concentration and carbon isotope ratio of methane ($\delta^{13}C_{CH4}$)

ii) Significant increase in DIC concentration and depletion carbon isotope ratio of DIC ($\delta^{13}C_{DIC}$)

iii) Increase in $Fe^{2+}$ and $Mn^{2+}$ concentration in porewater.

In our manuscript, we have shown all the above signatures which are unequivocal evidence for Fe-Mn AOM. We have also explained the possible influence of metal sulfide nanoparticles in the porewaters (response to review comment no.1) which possibly resulted in the high concentration of Fe in the porewaters. The nanoparticles form due to ferrous production in the porewaters by ferric reduction through AOM. We have also explained in response to comment no. 34 that Fe-Mn spikes are only restricted to Fe-Mn-AOM zones and not erratically distributed all over the cores. We believe that this is convincing evidence for Fe-Mn-AOM.

Comment 39: Line 245-246 and later. See earlier comments. $Fe^{2+}$ and $H_2S$ do not generally co-occur. You should assess your methods to exclude potential artifacts.

**Reply:** Please see response to review comment no:1.

Comment 40: Line 256 more solid phase data are needed e.g. S or FeS and FeS$_2$ data

**Reply**: S$^0$ and FeS were below the detection limit in the present study. The data of FeS$_2$ content is provided below (Figure 9D) which will be included in the revised text. No obvious enrichment in FeS$_2$ content is observed at Fe-Mn-AOM depths in Zone I, II, and III (Figure 9) to support the precipitation of FeS$_2$ from Fe$^{2+}$ produced via Fe-AOM. We have carried out a calculation to estimate the amount of Fe$^{2+}$ (produced via Fe-AOM) that might have reacted with H$_2$S to form pyrite. The observed exact correspondence of DIC concentration and AOM zones (depleted $\delta^{13}C_{DIC}$ and $\delta^{13}C_{CH4}$ values) suggests that DIC production is primarily controlled by AOM. Since these AOM zones are also accompanied by porewater Fe-Mn increase, we attribute the DIC enrichment primarily to Fe-Mn-AOM (Luo et al., 2020). The DIC concentration perturbations (along the blue dashed lines in Figure 3) may be converted to equivalent µM of Fe$^{2+}$ produced via Fe$^{3+}$ reduction in the pore waters. The stoichiometrically calculated amount of Fe$^{2+}$ produced at Fe-AOM depths are 17163.18, 11673.09, 16743.20, 12883.82, 16854.54, 8159.55, 6192.97, 7104.52 µM, which is equivalent to CRS contribution of 2.06, 1.4, 2.01, 1.54, 2.02, 0.97, 0.74, 0.85 mg/g. The measured bulk CRS content (Figure 9D) in those Fe-Mn-AOM peak depths corresponds to 19.66, 47.3, 20.33, 51.3, 57.7, 72.88, 50.98, 73.8 mg/g which is significantly high compared to CRS which might have formed from Fe produced via Fe-AOM. This may explain the lack of significant enrichment in FeS$_2$ at Fe-AOM depths. Moreover, since pyritization in sediments is a cumulative process throughout sediment diagenesis, primarily controlled by the rate of microbial sulfate reduction, sedimentation rate, labile organic flux (Berner, 1985; Raiswell and Berner, 1985; Wilkin and Barnes, 1997; Werne et al., 2003; Markovic et al., 2015), bottom water oxygenation which supports benthic fauna causing bioturbation and subsequent reoxidation of Fe-sulfide minerals (Chambers et al., 2000; Antler et al., 2019), and the availability of Fe that can react with sulfide (Jørgensen, 1982; Yucel et al., 2010; Zhu et al., 2016; Jørgensen et al., 2019), the FeS$_2$ content at specific Fe-AOM depths does not show any distinct concentration spikes. Both HS$^-$ concentration and isotope profile of HS$^-$ does not match with that of CRS in the present (Figure 9 C, E) and previous studies (For eg: Raven et al., 2016; Fernandes et al., 2020) which itself reflects the cumulative effect of sulfidization on speciation and isotope ratios during sediment burial. Therefore, given the large range in FeS$_2$ content, partitioning of H$_2$S in iron sulfide and organic bound sulfur (OBS) and the complexity involved in the cumulative nature of CRS and OBS makes quantification of Fe-AOM contribution towards additional FeS/CRS precipitation

a little challenging especially through gravimetry. We will include this aspect suggested by the reviewer in the revised manuscript.

[Figure]

*Figure 9: Depth profiles of porewater (A) $SO_4^{2-}$, $\Sigma HS^-$, (B) $Fe^{2+}$, (C) $\delta^{34}S_{SO4}{}^{2-}$, $\delta^{34}S_{\Sigma HS^-}$, (D) CRS and (E) $\delta^{34}S_{CRS}$ in SSD070/7/GC6 (present study).*

Comment 41: Line 256. Mn-sulfide formation is rare. Here and elsewhere, the manuscript would benefit from Mn speciation for the sediment.

**Reply:** Unfortunately, the $MnO_2$ speciation data of the core is not available. But we have bulk Mn data which is provided below. The bulk $MnO_2$ doesn't show any correlation with porewater $Mn^{2+}$ spikes.

[Figure]

*Figure 10: Porewater Mn²⁺ and solid phase bulk Mn concentration in the present study.*

Comment 42: Line 258. Change to "colloids".

**Reply:** Thank you for the suggestion. The collides will be changed to colloids.

Comment 43: Line 263: Oxidation artifacts often lead to erratic profiles so you cannot exclude artefacts here.

**Reply:** Please see the response to review comment no:1

Comment 44: Line 266. Convert to μmol/g.

**Reply:** Thank you for your suggestion. We will convert the concentration data to μmol/g in text and figures accordingly.

Comment 45: Line 275. Rephrase to "throughout the sediment depth assessed"

**Reply:** Thank you so much for the comment. The suggested change will be incorporated in the revised manuscript.

Comment 46: Line 286. See the comment above a plausible scenario with a timeline of deposition and diagenesis is needed to explain what the sediment in these 500 cm represents, the depositional regime, and how it has been altered upon burial.

**Reply:** The sedimentation took place in the hypoxic inner shelf off the Eastern Arabian Sea. Based on the previous $^{210}$Pb dating from the present coring site, the sedimentation rate ranges from 0.19 cm/yr (below 40 cmbsf) to 1.5 cm/yr (above 40 cmbsf) (Sebastian et al., 2017). The grain size analysis data shows a dominantly clayey silt sediment type where the clay minerals are composed of kaolinite and montmorillonite with very low illite content. The clay mineralogy composition in this region doesn't show any significant variation along the sediment core. Moreover, the zones are also homogenous in terms of porosity except for some spikes in between (Figure 4A). The median grain size (D50) shows only a small variability of 6.66 to 17.43 μm throughout the core (Figure 4D). However, the TOC content, TOC/TN$_{molar}$ ratio, and $\delta^{13}C_{TOC}$ values show significant variation throughout the sediment core indicating variability in the nature and composition of organic matter in the study area. It's important that the variation in TOC content, TOC/TN$_{(molar)}$, and $\delta^{13}C_{TOC}$ values do not show any obvious correlation with the porewater Fe-Mn spikes.

Comment 46: What about Mn oxides? Note that the TOC data belongs in the main manuscript

**Reply:** Unfortunately, the MnO$_2$ speciation data of the core is not available. But we have bulk Mn data which is provided in figure 10. As per the suggestion, the TOC data will be incorporated into the main text.

Comment 47: Line 299. The relevance of the delta $^{15}$N data and TOC/TON data to this paper is not clear.

**Reply:** The shallow shelf off Western Continental shelf of India (WCSI) is highly dynamic in terms of drastic changes in water column redox conditions, marine productivity and fluvial fluxes of organic matter, sediment load, and extensive denitrification (Naqvi et al., 2000; Schott et al., 2001; Naqvi et al., 2006; Maya et al., 2011; Mazumdar et al., 2012; Fernandes et al., 2020). The significant changes in sedimentary $\delta^{15}N$ observed in the present and previous studies (Agnihotri et al., 2009) indicate drastic changes in denitrification conditions in the water column off WCSI. Moreover, the TOC, TOC/TN and $\delta^{13}C_{TOC}$ data in the present and previous studies (Mazumdar et al., 2012; Fernandes et al., 2020) also show significant variation throughout the core indicating variability in the nature and composition of organic matter in the study area.

Comment 48: Line 304. Data on microorganisms in the water column cannot be directly coupled to those in the sediment

**Reply:** Previous studies indicated that after burial, microbial populations sourced from the overlying water column undergo selection while competing for energy-yielding substrates within the sediments, leading to the formation of a genetically distinct deep biosphere (Orsi et al., 2016). However, the study by Orsi et al. (2017) revealed that 5–15% of taxa in Arabian Sea sediments serve as indicators of past oceanographic conditions, suggesting they are subject to weaker selection pressures. Besides the availability of terminal electron acceptors for respiration, paleoenvironmental conditions may account for a portion of the stratigraphic microbial distributions in marine sediments (Orsi et al., 2017). We have only that suggested microbiological studies has to be conducted in future for a clearer understanding of sedimentary processes leading to Fe-Mn-AOM.

Therefore, we hypothesize a dominant role of the localized abundance of metal-reducing bacterial/archaeal communities in restricting Fe-Mn-AOM activities into specific layers. The focusing of bacterial activity in different sediment layers may be attributed to factors such as past environmental conditions/depositional processes (Orsi et al., 2017; Hoshino et al., 2020) and the nature of sedimentary material which in turn may be more pronounced in coastal regions subjected to intense climate change (seasonal variation) (Parkes et al., 2000; Orsi et al., 2017).

Comment 49: The conclusion on focusing of microbial communities in distinct layers is speculative.

**Reply:** We have not speculated anything in the conclusion. We have only suggested the possibility of microbial role in driving focussed Fe-Mn-AOM in the present study and may be investigated in future studies, which we will be carrying out in future expeditions. To gather additional porewater data, we have to carry out another research cruise which is not possible right now. We will be carrying out more higher resolution fluid chemistry, solid phase chemistry, and microbiology studies along with microbiology group in the future expeditions. Additional solid phase data (sedimentological characteristics) have been provided in Figure 4 and discussed in response to review comment 3(A) which also do not show any obvious correlation with the observed porewater profiles. **However, we will include this aspect suggested by the reviewer in the revised text.**

Comment 50: The metagenomic data for a core from a site in the vicinity cannot be used. The data do not support the conclusion that there are distinct layers with microbes that can carry out Fe-Mn-AOM

**Reply:** The metagenomic data for a nearby core may be removed from the revised manuscript.

Comment 51: Line 344-346: The data set is not comprehensive enough and does not provide evidence for Fe-Mn-AOM, see comments above.

**Reply:** Following are the evidence for Fe-Mn-AOM

Metal driven AOM are typically characterized by the following biogeochemical signatures (Riedinger et al., 2014; Egger et al., 2016a, b, 2017; Vigderovich et al., 2019; Li et al., 2019; Luo et al., 2020; Aromokeye et al., 2020; Xiao et al., 2023).

    i)       Significant depletion in methane concentration and carbon isotope ratio of methane ($\delta^{13}C_{CH4}$)

    ii)      Significant increase in DIC concentration and depletion carbon isotope ratio of DIC ($\delta^{13}C_{DIC}$)

    iii)     Increase in $Fe^{2+}$ and $Mn^{2+}$ concentration in porewater.

In our manuscript, we have shown all the above signatures which are unequivocal evidence for Fe-Mn AOM. We have also explained the possible influence of metal sulfide nanoparticles in the porewaters (response to review comment no.1) which possibly resulted in the high concentration of Fe in the porewaters. The nanoparticles form due to ferrous production in the porewaters by ferric reduction through AOM. We have also explained in response to comment no. 34 that Fe-Mn spikes are only restricted to Fe-Mn-AOM zones and not erratically distributed all over the cores. We believe that this is convincing evidence for Fe-Mn-AOM.

Comment 52: Line 347-348. Unless this is FeS that has passed the filter or an artifact, see above.

**Reply:** We deeply appreciate the reviewer for raising this very pertinent issue regarding iron sulfide nanoparticles in the porewater. We agree that we should have discussed this in the manuscript. At high hydrogen sulfide concentrations, $Fe^{2+}$ is likely to be present in the porewater as $FeS_{nano}$ (Matamoros-Veloza et al., 2018), mackinawite or any other stable Fe-S nanoparticle form (Rickard and Morse, 2005; Rickard and Luther, 2007). We believe that $Fe^{2+}$ is generated by $Fe^{3+}$ reduction via the AOM process. The $Fe^{2+}$ required for the formation of the

FeS nanoparticles may be produced via the Fe-AOM pathway. The nanoparticles will pass through 0.22-micron syringe filters and eventually be in the aliquots for metal concentration measurement. The FeS/MnS nanoparticles will dissolve in supra pure nitric acid and will be measured as the total Fe concentration in the pore water. It is worth noting that except for the Fe concentration spikes, the background Fe concentrations range from 9.05 to 43.89 μM compared to the spike values from 164.94 to 387.54 μM (Figure 3D). It is also apparent that the Fe spikes are strictly associated with the AOM zones identified by methane concentrations, methane carbon isotope ratios, DIC concentrations, and DIC carbon isotope ratios (Figure 3A-E). It may be noted that the $H_2S$ is high throughout the core below 63 cmbsf. Had it been a case of artifact generation, $Fe^{2+}$ spikes wouldn't have been restricted to the AOM zones.

**Comment 53: Line 349. The meaning of "Significant biogeochemical phenomenon" is not clear.**

**Reply:** We meant to convey that it is an important biogeochemical phenomenon in sulfate rich sediments.

**Comment 54: Line 358. The link with seasonal hypoxia is not clear**

**Reply:** The shallow shelf off WCSI is highly dynamic, experiencing drastic changes in water column redox conditions, marine productivity, fluvial fluxes of organic matter, sediment load, and extensive denitrification (Naqvi et al., 2000; Schott et al., 2001; Naqvi et al., 2006; Maya et al., 2011; Mazumdar et al., 2012; Fernandes et al., 2020). The silt-clay-rich sediments of the study area is characterized by TOC content ranging from 1.45 to 31.3 mg/g. Both $(TOC/TN)_{molar}$ ratios (3.47 to 27.32) and $\delta^{13}C_{TOC}$ values (-20.69 to -25.93 ‰) from the present and previous studies (Mazumdar et al., 2012; Fernandes et al., 2020) indicate marked temporal variation in the fluvial and marine organic matter fluxes in WCSI. Previous studies (near SSD070/7/GC6; Figure 1b) investigating carbon and nitrogen stable isotopes of suspended particulate organic matter (SPOM) in the estuary (Bardhan et al., 2014) and shelf zone (Maya et al., 2011) of WCSI revealed significant intra-annual variations in $\delta^{15}N$ (estuary: 0.69 to 7.26 ‰; shelf: -4.17 to 10.43 ‰) and $\delta^{13}C$ (estuary: -30.14 to -19.52 ‰, shelf: -17.64 to -26.74 ‰) throughout the year. These variations reflect the complex and dynamic nature of biogeochemical processes and organic matter sources in the coastal waters of the WCSI. Corroboratively, significant variations in the diversity, abundance, and activity of microorganisms, attributable to seasonal differences in nutrient availability, have been recorded in the water column between monsoon and non-monsoon seasons (Gomes et al., 2019; Naik et

al., 2024; Parab et al., *bioRxiv*). Spatiotemporally contrasting biogeochemical conditions of shallow coastal waters may have profound influence on the structure and function of underlying sedimentary microbiomes (Bhattacharya et al., 2021). However, in order to understand the influence of seasonal hypoxia on sediment biogeochemical processes, we have to subsample the sediment core at mm scale which is impossible.

Supplement:

Comment 55: Presentation of the TOC data in wt% would allow for more easy comparison to other published work.

**Reply:** The TOC content will be changed to wt %.

**References**

Agnihotri, R., Naqvi, S. W. A., Kurian, S., Altabet, M. A., and Bratton, J. F.: Is $d^{15}N$ of sedimentary organic matter a good proxy for paleodenitrification in coastal waters of the eastern Arabian Sea, Geophys. Monogr. Ser, 185, 321-332, 2009.

Alperin, M. J., Reeburgh, W. S., and Whiticar, M. J.: Carbon and hydrogen isotope fractionation resulting from anaerobic methane oxidation, Global biogeochemical cycles, 2, 279-288, 1988.

Antler, G., Mills, J. V., Hutchings, A. M., Redeker, K. R., & Turchyn, A. V. (2019). The sedimentary carbon-sulfur-iron interplay–A lesson from East Anglian salt marsh sediments. *Frontiers in Earth Science*, *7*, 140.

Aromokeye, D. A., Kulkarni, A. C., Elvert, M., Wegener, G., Henkel, S., Coffinet, S., Eickhorst, T., Oni, O. E., Richter-Heitmann, T., and Schnakenberg, A.: Rates and microbial players of iron-driven anaerobic oxidation of methane in methanic marine sediments, Frontiers in Microbiology, 10, 3041, https://doi.org/10.3389/fmicb.2019.03041, 2019.

Baikova, I. S., Shtamm, E. V., Vichutinskaya, E. V., and Skurlatov, Y. I.: The mechanism of oxidation of FeS nanoparticles by molecular oxygen and hydrogen peroxide in dilute aqueous solutions, Russian Journal of Physical Chemistry B, 3, 251-256, 2009.

Bardhan, P., Karapurkar, S. G., Shenoy, D. M., Kurian, S., Sarkar, A., Maya, M. V., Naik, H., Varik, S., and Naqvi, S. W. A.: Carbon and nitrogen isotopic composition of suspended

particulate organic matter in Zuari Estuary, west coast of India, Journal of Marine Systems, 141, 90-97, https://doi.org/10.1016/j.jmarsys.2014.07.009, 2014.

Berner, R. A.: Sulphate reduction, organic matter decomposition and pyrite formation, Philosophical Transactions of the Royal Society of London. Series A, Mathematical and Physical Sciences, 315, 25-38, 1985.

Behrouz, M. and Aghajani, M.: Solubility of methane, ethane, and propane in pure water using new binary interaction parameters, Iranian Journal of Oil and Gas Science and Technology, 4, 51-59, 2015.

Bhattacharya, S., Mapder, T., Fernandes, S., Roy, C., Sarkar, J., Rameez, M. J., Mandal, S., Sar, A., Chakraborty, A. K., and Mondal, N.: Sedimentation rate and organic matter dynamics shape microbiomes across a continental margin, Biogeosciences, 18, 5203-5222, https://doi.org/10.5194/bg-18-5203-2021, 2021.

Borowski, W. S., Paull, C. K., and Ussler III, W.: Carbon cycling within the upper methanogenic zone of continental rise sediments: An example from the methane-rich sediments overlying the Blake Ridge gas hydrate deposits., Marine Chemistry, 57, 299–311, https://doi.org/10.1016/S0304-4203(97)00019-4, 1997.

Borowski, W. S., Cagatay, N., Tournois, Y., and Paull, C. K.: Data report: Carbon isotopic composition of dissolved $CO_2$, $CO_2$ gas, and methane, Blake-Bahama Ridge and Northeast Bermuda Rise, ODP Leg 172, Proceedings ODP, Scientific Results, 172, 1, 2001.

Canfield, D.: Reactive iron in marine sediments, Geochimica et Cosmochimica Acta, 53, 619-632, https://doi.org/10.1016/0016-7037(89)90005-7, 1989.

Chambers, R. M., Hollibaugh, J., Snively, C. S., and Plant, J. N.: Iron, sulfur, and carbon diagenesis in sediments of Tomales Bay, California, Estuaries, 23, 1-9, 2000.

Chiriță, P., Descostes, M., & Schlegel, M. L.: Oxidation of FeS by oxygen-bearing acidic solutions, Journal of colloid and interface science, 321(1), 84-95, 2008.

Cho, H., Kim, S.-H., Shin, K.-H., Bahk, J.-J., and Hyun, J.-H.: Microbial Community Composition Associated with Anaerobic Oxidation of Methane in Gas Hydrate-Bearing Sediments in the Ulleung Basin, East Sea, The Sea: JOURNAL OF THE KOREAN SOCIETY OF OCEANOGRAPHY, 20, 53-62, 2015

Emeis, K. C., Brüchert, V., Currie, B., Endler, R., Ferdelman, T., Kiessling, A., Leipe, T., Noli-Peard, K., Struck, U., and Vogt, T.: Shallow gas in shelf sediments of the Namibian coastal upwelling ecosystem, Continental Shelf Research, 24, 627-642, 2004.

Egger, M., Rasigraf, O., Sapart, C. J., Jilbert, T., Jetten, M. S. M., Rockmann, T., Van der Veen, C., Banda, N., Kartal, B., and Ettwig, K. F.: Iron-mediated anaerobic oxidation of methane in brackish coastal sediments, Environmental science & technology, 49, 277-283, 2015.

Egger, M., Lenstra, W., Jong, D., Meysman, F. J. R., Sapart, C. l. J., Van der Veen, C., Röckmann, T., Gonzalez, S., and Slomp, C. P.: Rapid sediment accumulation results in high methane effluxes from coastal sediments, PloS one, 11, e0161609, 2016a.

Egger, M., Kraal, P., Jilbert, T., Sulu-Gambari, F., Sapart, C. J., Röckmann, T., and Slomp, C. P.: Anaerobic oxidation of methane alters sediment records of sulfur, iron and phosphorus in the Black Sea, Biogeosciences, 2016b.

Egger, M., Hagens, M., Sapart, C. l. J., Dijkstra, N., van Helmond, N. A. G. M., Mogollón, J. M., Risgaard-Petersen, N., van der Veen, C., Kasten, S., and Riedinger, N.: Iron oxide reduction in methane-rich deep Baltic Sea sediments, Geochimica et Cosmochimica Acta, 207, 256-276, 2017.

Fernandes, S., Mazumdar, A., Bhattacharya, S., Peketi, A., Mapder, T., Roy, R., Carvalho, M. A., Roy, C., Mahalakshmi, P., and Da Silva, R.: Enhanced carbon-sulfur cycling in the sediments of Arabian Sea oxygen minimum zone center, Scientific reports, 8, 8665, 2018.

Fernandes, S., Mazumdar, A., Peketi, A., Anand, S. S., Rengarajan, R., Jose, A., Manaskanya, A., Carvalho, M. A., and Shetty, D.: Sulfidization processes in seasonally hypoxic shelf sediments: a study off the West coast of India, Marine and Petroleum Geology, 117, 104353, 2020.

Geprägs, P., Torres, M. E., Mau, S., Kasten, S., Römer, M., and Bohrmann, G.: Carbon cycling fed by methane seepage at the shallow Cumberland Bay, South Georgia, sub-Antarctic, Geochemistry, Geophysics, Geosystems, 17, 1401-1418, 2016.

Gomes, J., Khandeparker, R., Meena, R. M., and Ramaiah, N.: Bacterial community composition markedly altered by coastal hypoxia, Journal of Microbiology, 59, 200-208, 2019.

Hensen, C., Zabel, M., Pfeifer, K., Schwenk, T., Kasten, S., Riedinger, N., Schulz, H., and Boetius, A.: Control of sulfate pore-water profiles by sedimentary events and the significance of anaerobic oxidation of methane for the burial of sulfur in marine sediments, Geochimica et Cosmochimica Acta, 67, 2631-2647, 2003.

Holler, T., Wegener, G., Niemann, H., Deusner, C., Ferdelman, T. G., Boetius, A., Brunner, B., and Widdel, F.: Carbon and sulfur back flux during anaerobic microbial oxidation of methane and coupled sulfate reduction, Proceedings of the National Academy of Sciences, 108, E1484-E1490, 2011.

Hori, T., Aoyagi, T., Itoh, H., Narihiro, T., Oikawa, A., Suzuki, K., Ogata, A., Friedrich, M. W., Conrad, R., and Kamagata, Y.: Isolation of microorganisms involved in reduction of crystalline iron (III) oxides in natural environments, Frontiers in Microbiology, 6, 386, 2015.

Hoshino, T., Doi, H., Uramoto, G.-I., Wörmer, L., Adhikari, R. R., Xiao, N., Morono, Y., D' Hondt, S., Hinrichs, K.-U., and Inagaki, F.: Global diversity of microbial communities in marine sediment,Proceedings of the National Academy of Sciences of the United States of America, 117, 27587-27597, 2020.

Hyacinthe, C., Bonneville, S., and Van Cappellen, P.: Reactive iron (III) in sediments: chemical versus microbial extractions, Geochimica et Cosmochimica Acta, 70, 4166-4180, https://doi.org/10.1016/j.gca.2006.05.018, 2006.

Jørgensen, B. B., Findlay, A. J., and Pellerin, A.: The biogeochemical sulfur cycle of marine sediments, Frontiers in Microbiology, 10, 849, 2019.

Jørgensen, B. B.: Mineralization of organic matter in the sea bed—the role of sulphate reduction, Nature, 296, 643-645, 1982.

Knab, N. J., Cragg, B. A., Hornibrook, E. R. C., Holmkvist, L., Pancost, R. D., Borowski, C., Parkes, R. J., and Jørgensen, B. B.: Regulation of anaerobic methane oxidation in sediments of the Black Sea, Biogeosciences, 6, 1505-1518, 2009.

Lalonde, K., Mucci, A., Ouellet, A., and Gélinas, Y.: Preservation of organic matter in sediments promoted by iron, Nature, 483, 198-200, 2012.

Li, Q., Wang, F., Chen, Z., Yin, X., and Xiao, X.: Stratified active archaeal communities in the sediments of Jiulong River estuary, China, Frontiers in Microbiology, 3, 311, 2012.

Li, J., Li, L., Bai, S., Ta, K., Xu, H., Chen, S., Pan, J., Li, M., Du, M., and Peng, X.: New insight into the biogeochemical cycling of methane, S and Fe above the Sulfate-Methane Transition Zone in methane hydrate-bearing sediments: A case study in the Dongsha area, South China Sea, Deep Sea Research Part I: Oceanographic Research Papers, 145, 97-108, 2019.

Lovley, D. R. and Phillips, E. J. P.: Organic matter mineralization with reduction of ferric iron in anaerobic sediments, Applied and environmental microbiology, 51, 683-689, 1986.

Lovley, D. R. and Phillips, E. J. P.: Novel mode of microbial energy metabolism: organic carbon oxidation coupled to dissimilatory reduction of iron or manganese, Applied and environmental microbiology, 54, 1472-1480, 1988.

Luo, M., Torres, M. E., Hong, W.-L., Pape, T., Fronzek, J., Kutterolf, S., Mountjoy, J. J., Orpin, A., Henkel, S., and Huhn, K.: Impact of iron release by volcanic ash alteration on carbon cycling in sediments of the northern Hikurangi margin, Earth and Planetary Science Letters, 541, 116288, 2020

Markovic, S., Paytan, A., and Wortmann, U. G.: Pleistocene sediment offloading and the global sulfur cycle, Biogeosciences, 12, 3043-3060, 2015.

Matamoros-Veloza, A., Cespedes, O., Johnson, B. R. G., Stawski, T. M., Terranova, U., de Leeuw, N. H., and Benning, L. G.: A highly reactive precursor in the iron sulfide system, Nature communications, 9, 3125, 2018.

Mehra, O. and Jackson, M.: Iron oxide removal from soils and clays by a dithionite-citrate system buffered with sodium bicarbonate, Clays and clay Minerals, 7, 317-327, 1960.

Maya, M. V., Karapurkar, S. G., Naik, H., Roy, R., Shenoy, D. M., and Naqvi, S. W. A.: Intra-annual variability of carbon and nitrogen stable isotopes in suspended organic matter in waters of the western continental shelf of India, Biogeosciences, 8, 3441-3456, 2011.

Mazumdar, A., Peketi, A., Joao, H., Dewangan, P., Borole, D. V., and Kocherla, M.: Sulfidization in a shallow coastal depositional setting: Diagenetic and palaeoclimatic implications, Chemical Geology, 322, 68-78, 2012.

Naqvi, S., Jayakumar, D., Narvekar, P., Naik, H., Sarma, V., D'Souza, W., Joseph, S., and George, M.: Increased marine production of N2O due to intensifying anoxia on the Indian continental shelf, Nature, 408, 346-349, 2000.

Naik, V., Damare, S. R., Shah, S. S., Shenoy, D., and Mulla, A. B.: Effect of coastal hypoxia on bacterial diversity as elucidated through 16S rRNA amplicon sequencing, Frontiers in Marine Science, 11, 1301955, 2024.

Norði, K. A., Thamdrup, B., and Schubert, C. J.: Anaerobic oxidation of methane in an iron-rich Danish freshwater lake sediment, Limnology and Oceanography, 58, 546-554, 2013.

Olson, L., Quinn, K. A., Siebecker, M. G., Luther Iii, G. W., Hastings, D., and Morford, J. L.: Trace metal diagenesis in sulfidic sediments: Insights from Chesapeake Bay, Chemical Geology, 452, 47-59, 2017.

Orsi, W. D., Barker Jørgensen, B., and Biddle, J. F.: Transcriptional analysis of sulfate reducing and chemolithoautotrophic sulfur oxidizing bacteria in the deep subseafloor, Environmental microbiology reports, 8, 452-460, 2016.

Orsi, W. D., Coolen, M. J. L., Wuchter, C., He, L., More, K. D., Irigoien, X., Chust, G., Johnson, C., Hemingway, J. D., and Lee, M.: Climate oscillations reflected within the microbiome of Arabian Sea sediments, Scientific Reports, 7, 6040, 2017

Parab, A. S., Ghose, M. P., and Manohar, C. S.: Variations in the bacterial community at Chlorophyll Maximum (C-Max) depths along the west coast of India due to seasonal changes in the primary productivity, bioRxiv [preprint], 2023.2007. 2005.547789, 2023.

Parkes, R. J., Cragg, B. A., and Wellsbury, P.: Recent studies on bacterial populations and processes in subseafloor sediments: a review, Hydrology Journal, 8, 11-28, 2000.

Peketi, A., Mazumdar, A., Joao, H. M., Patil, D. J., Usapkar, A., and Dewangan, P.: Coupled C-S-Fe geochemistry in a rapidly accumulating marine sedimentary system: diagenetic and depositional implications, Geochemistry, Geophysics, Geosystems, 16, 2865-2883, 2015.

Raiswell, R. and Berner, R. A.: Pyrite formation in euxinic and semi-euxinic sediments, American Journal of Science, 285, 710-724, 1985.

Raiswell, R., Vu, H. P., Brinza, L., and Benning, L. G.: The determination of labile Fe in ferrihydrite by ascorbic acid extraction: methodology, dissolution kinetics and loss of solubility with age and de-watering, Chemical Geology, 278, 70-79, 2010.

Raven, M. R., Sessions, A. L., Adkins, J. F., and Thunell, R. C.: Rapid organic matter sulfurization in sinking particles from the Cariaco Basin water column, Geochimica et Cosmochimica Acta, 190, 175-190, 2016.

Rickard, D. and Morse, J. W.: Acid volatile sulfide (AVS), Marine chemistry, 97, 141-197, 2005.

Rickard, D. and Luther, G. W.: Chemistry of Iron Sulfides, Chemical Reviews, 107, 514-562, 2007.

Riedinger, N., Formolo, M. J., Lyons, T. W., Henkel, S., Beck, A., and Kasten, S.: An inorganic geochemical argument for coupled anaerobic oxidation of methane and iron reduction in marine sediments, Geobiology, https://doi.org/10.1111/gbi.12077, 2014.

Sauer, S., Knies, J., Lepland, A., Chand, S., Eichinger, F., Schubert, C.J., 2015. Hydro-carbon sources of cold seeps off the Vesterålen coast, northern Norway. Chem. Geol.417, 371–382.

Sebastian, T., Nath, B. N., Naik, S., Borole, D. V., Pierre, S., and Yazing, A. K.: Offshore sediments record the history of onshore iron ore mining in Goa State, India, Marine pollution bulletin, 114, 805-815, 2017.

Sivan, O., Adler, M., Pearson, A., Gelman, F., Bar-Or, I., John, S. G., and Eckert, W.: Geochemical evidence for iron-mediated anaerobic oxidation of methane, Limnology and Oceanography, 56, 1536-1544, 2011.

Schott, F. A. and McCreary Jr, J. P.: The monsoon circulation of the Indian Ocean, Progress in Oceanography, 51, 2001.

Segarra, K. E. A., Comerford, C., Slaughter, J., and Joye, S. B.: Impact of electron acceptor availability on the anaerobic oxidation of methane in coastal freshwater and brackish wetland sediments, Geochimica et Cosmochimica Acta, 115, 15-30, 2013.

Treude, T., Krause, S., Maltby, J., Dale, A. W., Coffin, R., and Hamdan, L. J.: Sulfate reduction and methane oxidation activity below the sulfate-methane transition zone in Alaskan Beaufort Sea continental margin sediments: Implications for deep sulfur cycling, Geochimica et Cosmochimica Acta, 144, 217-237, 2014.

Vigderovich, H., Liang, L., Herut, B., Wang, F., Wurgaft, E., Rubin-Blum, M., and Sivan, O.: Evidence for microbial iron reduction in the methanic sediments of the oligotrophic southeastern Mediterranean continental shelf, Biogeosciences, 16, 3165-3181, 2019.

Wallace, P. J., Dickens, G. R., Paull, C. K., & Ussler, W.: Effects of core retrieval and degassing on the carbon isotope composition of methane in gas hydrate-and free gas-bearing sediments from the Blake Ridge, In Proceedings of the ocean drilling program. Scientific Results, 164, 101-112, 2000.

Wegener, G., Gropp, J., Taubner, H., Halevy, I., and Elvert, M.: Sulfate-dependent reversibility of intracellular reactions explains the opposing isotope effects in the anaerobic oxidation of methane, Science Advances, 7, eabe4939, 2021.

Werne, J. P., Lyons, T. W., Hollander, D. J., Formolo, M. J., & Damsté, J. S. S.: Reduced sulfur in euxinic sediments of the Cariaco Basin: sulfur isotope constraints on organic sulfur formation, Chemical Geology, 195(1-4), 159-179, 2003.

Whiticar, M. J. Carbon and hydrogen isotope systematics of bacterial formation and oxidation of methane, Chemical Geology, 161, 291-314, 1999.

Wilkin, R. T., & Barnes, H. L:. Formation processes of framboidal pyrite. *Geochimica et Cosmochimica Acta*, 61(2), 323-339, 1997.

Xiao, X., Luo, M., Zhang, C., Zhang, T., Yin, X., Wu, X., Zhao, J., Tao, J., Chen, Z., and Liang, Q.: Metal-driven anaerobic oxidation of methane as an important methane sink in methanic cold seep sediments, Microbiology Spectrum, 11, e05337-05322, 2023.

Yang, X., Zhang, Y., Sun, X., Xu, L., & Chen, T.: Marine sediment nitrogen isotopes and their implications for the nitrogen cycle in the sulfate-methane transition zone. Frontiers in Marine Science, 9, 1101599, 2023.

Yoshinaga, M. Y., Holler, T., Goldhammer, T., Wegener, G., Pohlman, J. W., Brunner, B., Kuypers, M. M. M., Hinrichs, K.-U., and Elvert, M.: Carbon isotope equilibration during sulphate-limited anaerobic oxidation of methane, Nature Geoscience, 7, 190-194, 2014.

Yücel, M., Konovalov, S. K., Moore, T. S., Janzen, C. P., & Luther III, G. W.: Sulfur speciation in the upper Black Sea sediments, Chemical Geology, 269, 364-375, 2010.

Zhu, M. X., Chen, K. K., Yang, G. P., Fan, D. J., and Li, T.: Sulfur and iron diagenesis in temperate unsteady sediments of the East China Sea inner shelf and a comparison with tropical mobile mud belts (MMBs), Journal of Geophysical Research: Biogeosciences, 121, 2811-2828, 2016